# Zero-dimensional halide hybrid bulk glass exhibiting reversible photochromic ultralong phosphorescence

Fei Nie[1] & Dongpeng Yan [1] ✉

Dynamically responsive materials, capable of reversible changes in color appearance and/or photoemission upon external stimuli, have attracted substantial attention across various fields. This study presents an effective approach wherein switchable modulation of photochromism and ultralong phosphorescence can be achieved simultaneously in a zero-dimensional organic-inorganic halide hybrid glass doped with 4,4´-bipyridine. The facile fabrication of large-scale glasses is accomplished through a combined grinding-melting-quenching process. The persistent luminescence can be regulated through the photochromic switch induced by photo-generated radicals. Furthermore, the incorporation of the aggregation-induced chirality effect generates intriguing circularly polarized luminescence, with an optical dissymmetry factor ($g_{lum}$) reaching the order of $10^{-2}$. Exploiting the dynamic ultralong phosphorescence, this work further achieves promising applications, such as three-dimensional optical storage, rewritable photo-patterning, and multi-mode anti-counterfeiting with ease. Therefore, this study introduces a smart hybrid glass platform as a new photo-responsive switchable system, offering versatility for a wide array of photonic applications.

Smart luminescent materials, exhibiting reversible responses to external stimuli (such as light, temperature, pressure, and mechanics), have garnered significant attention due to their advanced photofunctional and optoelectronic applications[1–3]. Recently, there has been a noteworthy focus on dynamic room-temperature phosphorescence (RTP) featuring ultralong excited states[4–6]. Particularly, the photo-responsive attributes, offering tunable emissive colors and lifetimes, present revolutionary potential in multilevel encryption and optical storage[7–9]. For instance, information storage designs leveraging metal−organic frameworks crystals have demonstrated alterations in color, fluorescence, and RTP in response to irradiation stimuli[10]. To date, photo-controllable RTP materials have predominantly existed in the forms of single crystals, polymers, and powders (Supplementary Table 1). However, their inherent brittleness and processing challenges significantly hinder their ability to meet diverse requirements for photonic applications and devices.

Glassy compounds, representing a substantial materials family in our society, offer distinct advantages over the aforementioned forms. Notably, they feature simplicity in fabrication, high transparency and hardness, making them highly appealing in photonic fields[11–13]. In general, glasses with dense three-dimensional (3D) networks maintain a rigid and confined environment, inhibiting non-radiative relaxation of triplet excitons for potential RTP enhancement[13]. Furthermore, bulk glasses may serve as prospective host matrixes for doping of various photoactive components, such as quantum dots, rare-earth ions and fluorescence dyes[14–16]. Despite recent exploration of photochromic germanium borate glasses for optical storage[17], the development of large-scale photochromic glasses with tunable RTP emissions represents an untapped domain and a challenging goal.

Crystalline organic-inorganic metal halides (OIMHs) have garnered worldwide attention in the fields of solar cells, light-emitting

[1]Beijing Key Laboratory of Energy Conversion and Storage Materials, and Key Laboratory of Radiopharmaceuticals, Ministry of Education, College of Chemistry, Beijing Normal University, Beijing 100875, P. R. China. ✉e-mail: yandp@bnu.edu.cn

devices, and photodetectors[18–20]. However, limited efforts have been directed towards the fabrication of glassy OIMHs due to their relatively weak glass formation ability. This is attributed to the easy dissociation of organic components prior to the melting of OIMHs using the mainstream melt-quenched approach, alongside a pronounced tendency for crystallization upon cooling[21]. Moreover, the dynamic photofunctional tunability of OIMHs glasses falls short of meeting practical applications. In this context, we present a facile approach to fabricate bulk hybrid glassy OIMHs (Fig. 1), demonstrating simultaneous photochromism and dynamic photoresponsive RTP for the first time. The zero-dimensional (0D) hybrid OIMHs glass, denoted as P-Zn, features a well-defined $A_2BX_4$ composition (A = P+ ((methoxymethyl)triphenylphosphonium), B = $Zn^{2+}$, and X = $Cl^-$), and can be prepared through a convenient crystal-melting-quenching or grinding-melting-quenching procedure. Doping with 4,4′-bipyridine (BP) as an electron acceptor results in the OIMHs hybrid glass (P-Zn-BP) displaying photo-regulated RTP through reversible photochromism (Fig. 1a–c). Post-photo-stimulation, the transparent P-Zn-BP glass undergoes a recognizable color change from colorless to dark blue, accompanied by a gradual weakening of the bright green afterglow. The switchable RTP on-off and coloration-decoloration processes can be continuously recycled during irradiation-heating/dark treatment. Moreover, the P+-based glasses exhibit both RTP and color-tunable luminescence upon doping with different ions (Fig. 1d). Notably, circularly polarized luminescence (CPL) is observed in the glasses due to the aggregation-induced chirality effect, a phenomenon rarely explored in current glassy materials. The fast response of dynamic RTP and photochromic characteristics lends them to the development of 3D optical storage, reversible photo-patterning, and even sunglass protection. Therefore, this work not only provides a facile method for the fabrication of large-scale OIMHs glasses, but also represents the inaugural instance of a photochromic glass exhibiting reversible photo-modulated afterglow and chiral emission towards advanced photonic applications.

## Results

### Preparation of P-Zn OIMHs-based glasses

Through a gradual evaporation process at room temperature (R.T.), P-Zn single crystals can be grown from water-ethanol mixed solution containing (methoxymethyl)triphenylphosphonium chloride (P-Cl) and $ZnCl_2$ in a 2:1 molar ratio. The resulting P-Zn crystal crystallizes in the $C2/c$ space group of the monoclinic crystal system (Supplementary Table 2 and Supplementary Data 1). In its 0D structure, one $Zn^{2+}$ ion coordinates with four $Cl^-$ anions to form a tetrahedral $ZnCl_4^{2-}$ cluster, which engages in abundant hydrogen bonding interactions with the organic cations, such as C-H···Cl, at distances of 2.651 Å, 2.781 Å, and 2.814 Å (Fig. 2a, and Supplementary Figs. 1 and 2). The production of P-Zn hybrid glass can be achieved by heating the single crystals at 140 °C, followed by vitrification of crystalline OIMHs through a direct quenching process. Alternatively, a faster method involves grinding the mixture of P-Cl and $ZnCl_2$, followed by continuous melting and quenching. The preparation of other P+-based glasses follows the same procedure as that of the pristine P-Zn glass, resulting in analogous zinc-bromide/iodide (named P-Zn-Br and P-Zn-I), antimony-doped zinc-chloride (named P-Zn-Sb), antimony-chloride (named P-Sb), and P-Zn-BP systems (details in Supplementary Information).

### Structural characterizations

Temperature-dependent in situ powder X-ray diffraction (PXRD) was employed to investigate the crystal-glass transition of P-Zn (Fig. 2b). In the crystalline sample, several diffraction peaks gradually vanish as the temperature rises from 25 to 100 °C. Upon reaching 140 °C, a broad "hump" band appears in the PXRD pattern, signifying the formation of an amorphous liquid. This amorphous pattern persists after quenching to R.T., indicating structural disordering in the glassy sample. The amorphous nature is further confirmed by PXRD examination of P-Zn, P-Zn-BP, and other as-synthesized glasses at R.T. (Supplementary Fig. 3). Notably, crystalline P-Zn possesses a lower density than its glassy counterpart (1.402 vs. 1.711 g cm$^{-3}$), suggesting a tighter molecular packing in the glass sample[22].

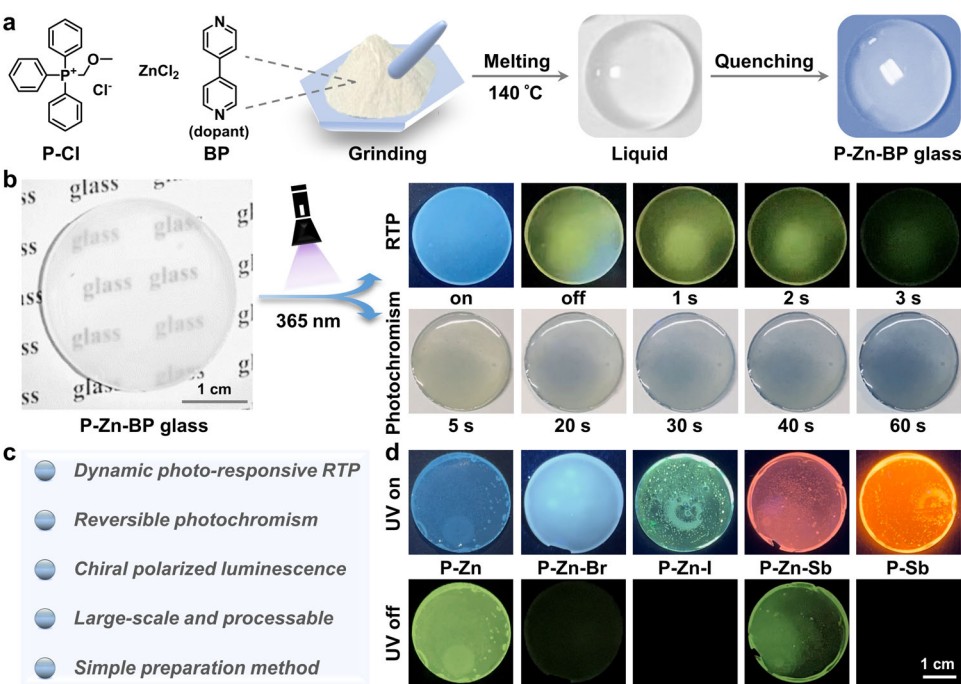

**Fig. 1 | Preparation and luminescent properties of the glasses. a** The preparation process of P-Zn-BP glass by using P-Cl, $ZnCl_2$ and BP (molar ratio: 2:1:0.02). **b** Ultralong room-temperature phosphorescence (RTP) and photochromic behaviors of P-Zn-BP glass under 365 nm irradiation at varying time intervals. **c** The typical feature of the BP-doped OIMHs glass. **d** Photographs of P-Zn and ion-doped glasses captured before and after cessation of 365 nm UV irradiation. The photographs were captured using an iPhone.

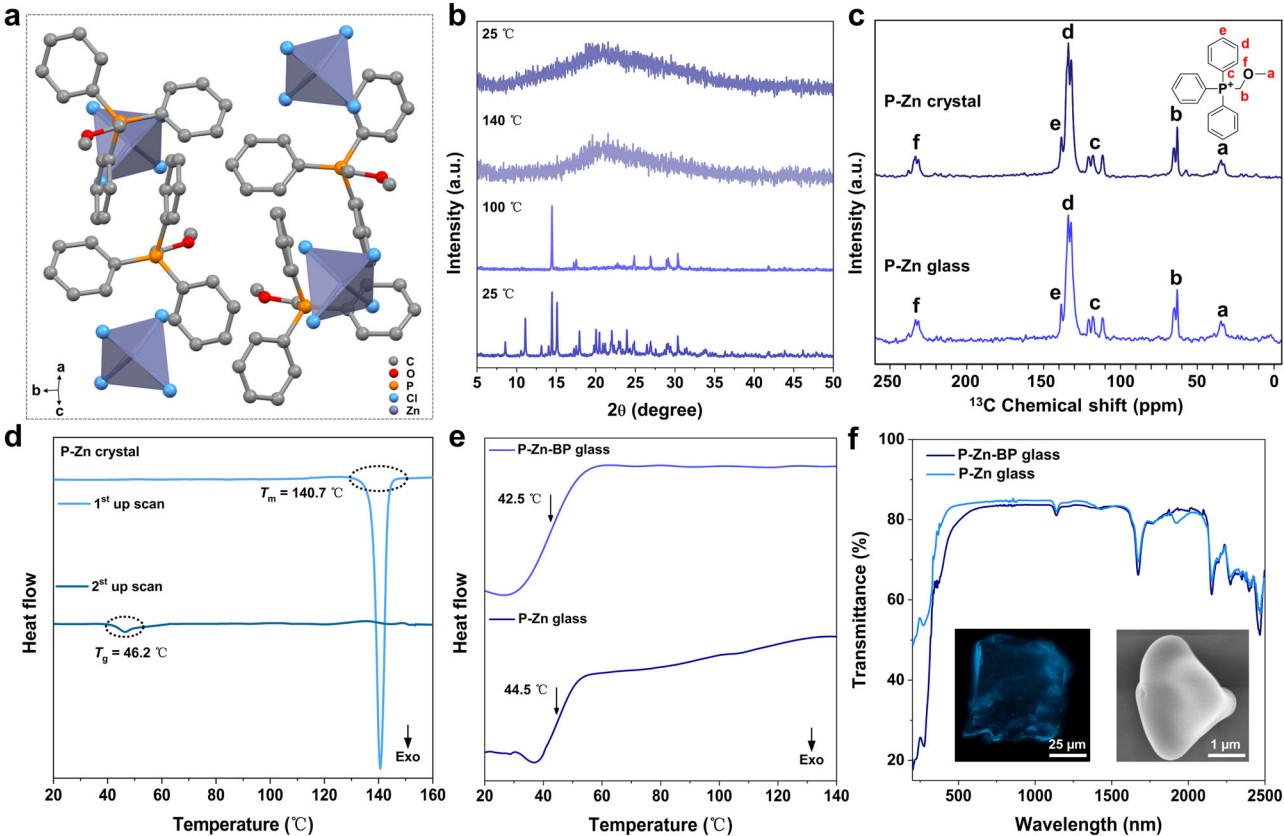

**Fig. 2 | The structure of the P-Zn OIMHs, and constitution and thermal properties of the transparent glasses. a** Simplified depiction of the P-Zn crystal structure, with omitted hydrogen atoms. **b** In situ PXRD measurements of P-Zn crystal at temperatures ranging from 25 to 140 °C, followed by a return to 25 °C. **c** Schematic illustration of carbon atoms labeled in the solid-state $^{13}$C NMR spectra of P-Zn glass and P-Zn crystal. **d** DSC curves of P-Zn crystal. **e** DSC curves of P-Zn and P-Zn-BP glasses. **f** UV-Vis-NIR transmittance spectra of P-Zn and P-Zn-BP glasses. The insets display fluorescence microscopy (left) and SEM images (right) of P-Zn-BP glass. a.u., arbitrary units.

The structures of the targeted OIMHs glasses were subsequently examined using solid-state $^{13}$C nuclear magnetic resonance (NMR), high-resolution electrospray ionization mass (HR-ESI-MS) spectrometry, and Fourier-transform infrared (FT-IR) spectroscopy. $^{13}$C magic angle spinning (MAS) NMR spectra of both crystalline and glassy P-Zn, collected under identical testing conditions (Fig. 2c), reveal similar chemical shifts. This confirms the maintenance of chemical environments and intermolecular interactions during the melting-quenching process[23]. In contrast to the crystalline sample, glassy P-Zn exhibits relatively poorly resolved NMR resonances, attributed to the disordered structure of the glass. HR-ESI-MS spectrometry for P-Zn glass and P-Zn-BP glass (both before and after irradiation) indicates a peak with $m/z = 307.1254$, 307.1253, and 307.1250, respectively, suggesting that BP doping or radiation has a minimal impact on the primary composition of OIMHs glasses (Supplementary Figs. 4–6). Comparison of the FT-IR spectra of P-Zn and BP-doped glasses reveals new peaks at 1486 and 3040 cm$^{-1}$, corresponding to the vibration of the pyridine ring and C-H of BP, respectively (Supplementary Figs. 7, 8)[24,25], implying the structural integrity of BP molecules in the doped glass.

Differential scanning calorimetry (DSC) testing was employed to elucidate the crystal-glass transformation. As depicted in Fig. 2d, during the initial up-scan, a notable endothermal peak at 140.7 °C emerges, denoted as the melting temperature ($T_m$), indicating the transition from a crystalline to a molten state. The point with the maximum slope of the capacity step at 46.2 °C in the subsequent up-scan, recognized as the glass transition temperature ($T_g$), confirms the formation of the glassy state for P-Zn. Thermogravimetric analysis (TGA) reveals that the decomposition temperature ($T_d$, approximately

300 °C) of P-Zn crystal significantly surpasses the $T_m$ (140.7 °C) (Supplementary Fig. 9a), a prerequisite for the preparation of melt-quenched glass[26]. In addition, a steplike transition, identified as the $T_g$, is observed on the DSC thermogram for P-Zn or P-Zn-BP glass (Fig. 2e), confirming their glassy states. Notably, evident $T_g$ points also appear in the DSC curves for analogous P-Zn-Br, P-Zn-I, P-Zn-Sb, and P-Sb systems (Supplementary Fig. 10), indicating the favorable glass-forming ability of P$^+$-based OIMHs. To the best of our knowledge, there have been very few reported instances of glassy OIMHs prepared via melting transitions. This rarity is primarily ascribed to the tendency of most OIMHs to decompose prior to melting, due to the thermal instability of their organic components[21]. In this case, the formation of a series of OIMHs glasses can be attributed to the meticulous selection of organic constituents, ensuring relatively high $T_d$ and low $T_m$ of the OIMHs. Specifically, the high stability ($T_d$, approximately 300 °C) of organic P$^+$ cations contributes to the elevated $T_d$ of the OIMHs (Supplementary Fig. 9b). Moreover, the incorporation of P$^+$ cations with large molecular sizes and tetrahedral clusters of ZnCl$_4^{2-}$ units may prevent ordering during the quenching process[27].

Ultraviolet-visible near-infrared (UV-vis-NIR) transmittance spectra reveal that P-Zn and BP-doped glasses exhibit approximately 80% transmittance in the 500–1500 nm range (Fig. 2f), a similar level observed in Sb-based glasses (Supplementary Fig. 11). Fluorescence microscopy and scanning electron microscope (SEM) images of the typical P-Zn-BP glass illustrate a high transparency and smooth surface (Fig. 2f). Elemental distribution mapping of SEM, including C, N, O, Zn, and Cl, depicts a homogeneous distribution of elements (Supplementary Fig. 12).

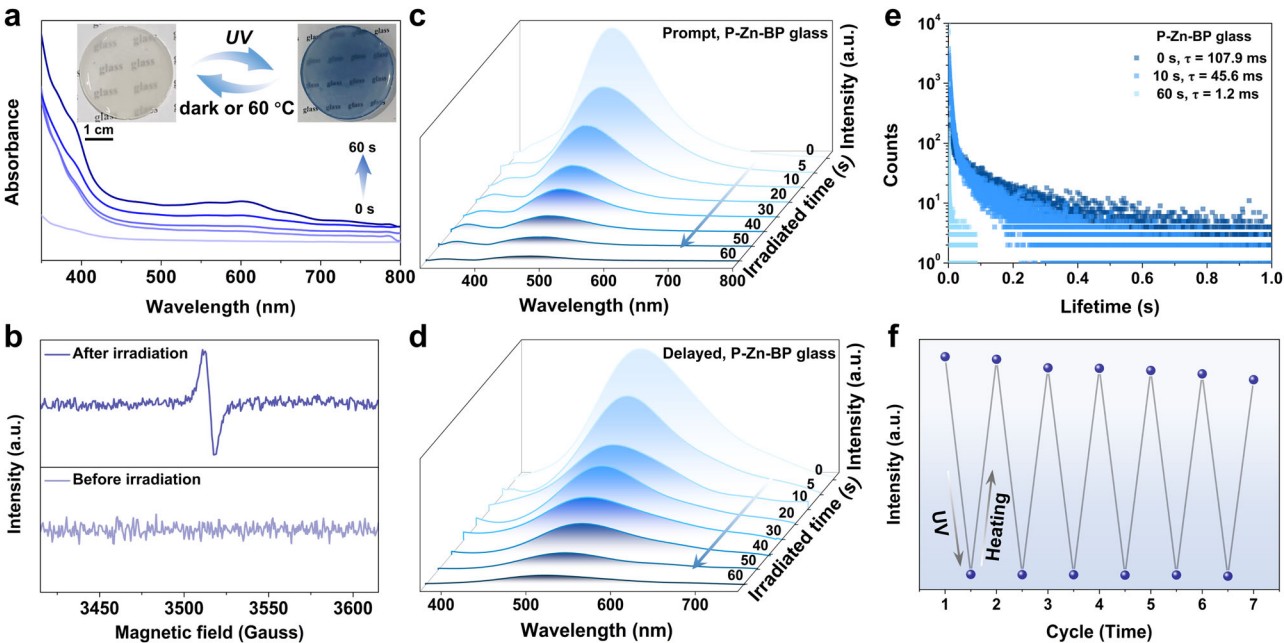

**Fig. 3 | Photophysical properties of the BP-doped glass. a** UV-vis spectra of P-Zn-BP glass at irradiation times of 0, 10, 20, 40, and 60 s. Inset: photographs of P-Zn-BP glass before and after 365 nm UV light irradiations. **b** EPR spectra of P-Zn-BP glass before and after 365 nm light irradiation. **c** Steady-state and **d** delayed emission spectra of P-Zn-BP glass at different irradiation times. **e** Long-lived RTP lifetime decay profiles of P-Zn-BP glass at 525 nm for various irradiation times. **f** Phosphorescence intensity (at 525 nm) of P-Zn-BP glass after alternating exposure to UV light and heating. a.u., arbitrary units.

High-resolution transmission electron microscopy (TEM) images and the selected-area electron diffraction (SAED) patterns of P-Zn and BP-doped glasses show the absence of lattice fringes (Supplementary Fig. 13), further implying their amorphous nature[28,29]. Nanoindentation measurements of bulk BP-doped glass yield an average Young's modulus of 9.52 GPa (Supplementary Fig. 14), comparable to those of reported hybrid glasses with cross-linked structures[30], emphasizing the structural rigidity induced by strong non-bonded interactions (such as hydrogen bonding and van der Waals forces). The uniform transparent appearance and high hardness make OIMHs glasses particularly appealing in photofunctional fields.

**Reversible photochromic ultralong phosphorescence**

The photoluminescent (PL) characteristics of the pristine P-Zn glass were initially investigated (Supplementary Fig. 15). P-Zn glass manifests prompt and delayed emissions at 472 and 542 nm, respectively, with an ultralong RTP lifetime of 124.0 ms. Upon UV excitation, visible blue emission and green afterglow are distinctly observable (Fig. 1d). In contrast, crystalline P-Zn exhibits a notably weaker RTP emission with a mere 6.2 μs lifetime (Supplementary Fig. 16). Notably, the transition from crystalline to glassy states enhances the RTP lifetime by five orders of magnitude (Supplementary Fig. 17). P-Cl powder shows photoemission akin to both P-Zn crystal and glass but with a shorter RTP lifetime of 3.3 μs (Supplementary Figs. 18 and 19), indicating the primary origin of RTP in P-Zn glass from the organic component (Supplementary Fig. 20). The significantly prolonged RTP lifetime of P-Zn glass (124.0 ms) compared with that of P-Zn crystal (6.2 μs) and P-Cl powder (3.3 μs) is attributed to the heavy atom effects of Zn and Cl ions, which effectively promote the intersystem crossing process. In addition, the enhanced density and stiffness of the glass matrix contribute to the formation of a more rigid structure, thus effectively suppressing non-radiative transitions of triplet excitons[13,31]. Doping other ions into P-Zn glass yields color-tunable luminescence, emphasizing the versatility of P-Zn as a glass fabrication platform for diverse photoemission. Particularly, P-Zn-Sb and P-Sb glasses exhibit unique near-infrared (NIR) emissions originating from self-trapped excitons in

hybrid Sb-containing halides (Supplementary Figs. 21–23)[32]. P-Zn-Br and P-Zn-I glasses display emission bands similar to P-Zn glass but with shortened RTP lifetimes (23.5 and 3.4 ms), ascribed to enhanced heavy atom effects via $Br^-$/$I^-$ compared to $Cl^-$ (Supplementary Figs. 24 and 25).

For P-Zn-BP glass, the phosphorescent lifetime (107.9 ms) is marginally shorter than that of pristine P-Zn, while the afterglow persists under ambient conditions (Fig. 3e). Interestingly, P-Zn-BP glass exhibits distinctive photo-responsive behavior: under UV irradiation at 365 nm for 60 s, the color shifts dramatically from colorless to dark blue. The photochromic process is reversible, with the color gradually reverting to its original state after 10 h in the dark or heating at 60 °C for 3 min in air (Fig. 3a). Beyond responsiveness to UV light (295–395 nm), P-Zn-BP glass demonstrates high sensitivity to a 300 W Xe lamp, X-ray, and even sunlight (Supplementary Fig. 26), highlighting the potential applications in detecting various types of radiation for human protection. Compared to state-of-the-art photochromic materials (Supplementary Table 1), P-Zn-BP glass exhibits relatively short photo-responsive time (60 s) and fast recovery time (3 min), while maintaining a long RTP lifetime (107.9 ms). These features underscore the exceptional dynamic ultralong phosphorescence in the large-scale OIMHs glass. Notably, other glasses without BP doping show no color change under the same irradiation conditions, further indicating the photochromism of P-Zn-BP glass is intricately linked to the incorporation of BP molecules.

The photochromism inherent in BP-doped glass was subjected to comprehensive analysis through UV-vis absorption and electron paramagnetic resonance (EPR) spectra. Upon exposure to 365 nm UV light, the UV-vis spectra exhibit the emergence of two distinct absorption bands at approximately 400 and 600 nm. The absorption intensity within the 350–700 nm range progressively increases with prolonged irradiation time, as illustrated in Fig. 3a. Concomitantly, the EPR spectrum shows a conspicuous signal characterized by a *g* value of 2.0016 after irradiation, while no signal is discernible prior to exposure (Fig. 3b). This observation underscores the pivotal role of light stimulation in generating BP radicals[33–35],

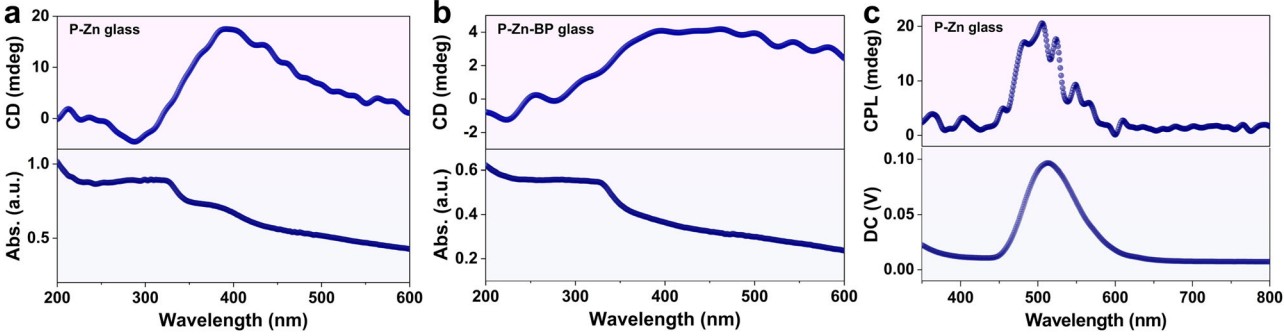

**Fig. 4 | Chiroptical properties of the glasses.** CD absorption spectra of **a** P-Zn and **b** P-Zn-BP glasses. **c** CPL spectrum of P-Zn glass. a.u., arbitrary units.

a hypothesis further substantiated through theoretical calculations (Supplementary Fig. 30).

The reversible photochromic nature of BP-doped glass proves instrumental in modulating PL behaviors. The irradiation time's augmentation results in intensity attenuation in both prompt (at approximately 300 and 510 nm) and delayed emissions (at approximately 525 nm) at R.T. (Fig. 3c, d, and Supplementary Fig. 27). Prior to coloration, observable green persistent luminescence lasts for a duration to the naked eye. However, following a 60-second irradiation period, the RTP lifetime experiences a substantial decrease from 107.9 to 1.2 ms (Fig. 3e). Temperature-dependent PL spectra measurements corroborate this, indicating a systematic decrease of intensity with increasing temperature from 77 to 297 K (Supplementary Fig. 28). This supports the assertion that the persistent emission at R.T. is attributed to ultralong phosphorescence, eliminating the possibility of thermally activated delayed fluorescence. Crucially, the reversible RTP switching is demonstrated to be repeatable for at least six cycles (Fig. 3f), underscoring the feasibility of modulating persistent luminescence in the bulk transparent glass through the photochromic process. It should be noted that after six cycles of repeated RTP switching, the coloration somewhat diminishes under the same irradiation conditions. This is because subjecting BP-doped glass to repeated photo-irradiation in ambient air may enhance the oxidative degradation of the photogenerated radical products[36].

Further investigation of the generation mechanism for tunable RTP and color in BP-doped glass elucidates a significant overlap between ultralong RTP emission and the absorption band spanning from 450 to 700 nm (Fig. 3a, d). This overlap suggests the potential occurrence of self-absorption in the photogenerated radical product. As the photoirradiation time increases, the absorption intensity at 600 nm gradually decreases, indicating that heightened levels of self-absorption predominantly contribute to the decrease of RTP intensity and lifetime post-coloration (Fig. 3d, e). Moreover, there is a blue shift in RTP luminescence from 523 to 514 nm as the exposure time to light increases. These results indicate that alterations in chemical structure and absorbance induced by photo-irradiation facilitate the realization of the photo-stimuli reversible RTP (Supplementary Fig. 29)[35].

Based on density functional theory (DFT) calculations (Supplementary Data 2), distribution maps of hole and electron transitions from the ground state ($S_0$) to the singlet state ($S_n$) in the P-Zn-BP model reveal evident electron transfer between electron donor Cl anions and electron acceptor BP molecules (Supplementary Figs. 30 and 31). The electron transfer process induces the formation of BP radicals, providing a competitive pathway for radiative transition in the form of phosphorescence. Notably, the powder obtained from grinding raw materials (P-Cl, $ZnCl_2$, and BP) does not exhibit a discernible photochromic response under 365 nm lamp irradiation (Supplementary Fig. 32). Substituting Zn with Sb cations, or Cl with Br anions (characterized by relatively weak electron-donating ability), results in different responses: the former manifests a discernible color change

upon 365 nm lamp irradiation, while the latter does not exhibit this characteristic. Moreover, doping of BP into traditional polymer films such as polyvinyl alcohol, polyvinylpyrrolidone, and gelatin—comprising mainly C, H, O, and N atoms—fails to induce any photochromic behaviors. This deficiency may stem from the absence of suitable units for electron donation within these polymers. These findings underscore the critical role of OIMHs glass formation and composition in fabricating stimuli-responsive materials.

## Chiro-optical properties

Recently, chiral OIMHs have emerged as pivotal systems in the realm of chiroptoelectronics, finding applications in photodetectors, circularly polarized light-emitting devices, 3D displays, and spintronics[37–39]. Circular dichroism (CD) and CPL of OIMHs are predominantly induced by the transfer of chiral information from organic to inorganic components through the incorporation of chiral organic molecules. Nevertheless, the origin of chiroptical properties in OIMHs might extend beyond mere "chirality transfer," as chiral features have been observed in amorphous films and powders even in the absence of chiral molecules[40]. For instance, certain propeller-like aggregation-induced emission (AIE) molecules devoid of intrinsic chiral units can exhibit latent chirality resulting from the disruption of mirror-image symmetry[41]. Notably, achiral tetraphenylethylene (TPE) and its derivatives demonstrate aggregation-induced chirality in their condensed phases[42–44]. In this study, $P^+$ adopts a propeller-like configuration due to steric repulsion between its multiple phenyl rings, resulting in pronounced AIE phenomena in P-Zn in EtOH-$H_2O$ suspensions (Supplementary Figs. 33 and 34), and the manifestation of chirality features in both the suspensions and the solid state distributed in the KBr pallet (Supplementary Fig. 35). However, no CD signal is detectable in the solution state due to rapid and reversible conformational changes. Thus, the P-Zn OIMHs can be characterized as a typical prochiral AIE system, where its chirality is induced when its three phenyl rings are fixed in a preferred clockwise or anticlockwise orientation in the aggregated state, similar to well-studied derivatives based on TPE[42–44], tetraphenylpyrazine[45], cyclooctatetrathiophene[46], and hexaphenylsilole[47].

Considering that the hybrid glasses were composed of $P^+$-based OIMHs, we delved into their chiro-optical characteristics. Remarkably, the CD spectra exhibit substantial responses in both P-Zn and BP-doped glasses (Fig. 4a, b). Furthermore, P-Zn glass displays notable CPL, encompassing both fluorescence and phosphorescence emissions with a peak at approximately 505 nm (Fig. 4c), intrinsically aligned with its prompt PL spectrum (Supplementary Fig. 15a). Conversely, BP-doped glass exhibits no appreciable CPL, due to the occurrence of photochromism under high-power 320 nm laser irradiation. NIR CPL emissions spanning from 400 to 900 nm, with peaks at approximately 650 nm, are detected for both P-Zn-Sb and P-Sb glasses (Supplementary Figs. 36 and 37), consistent with their prompt PL spectra (Supplementary Figs. 22a and 23a). This unequivocally

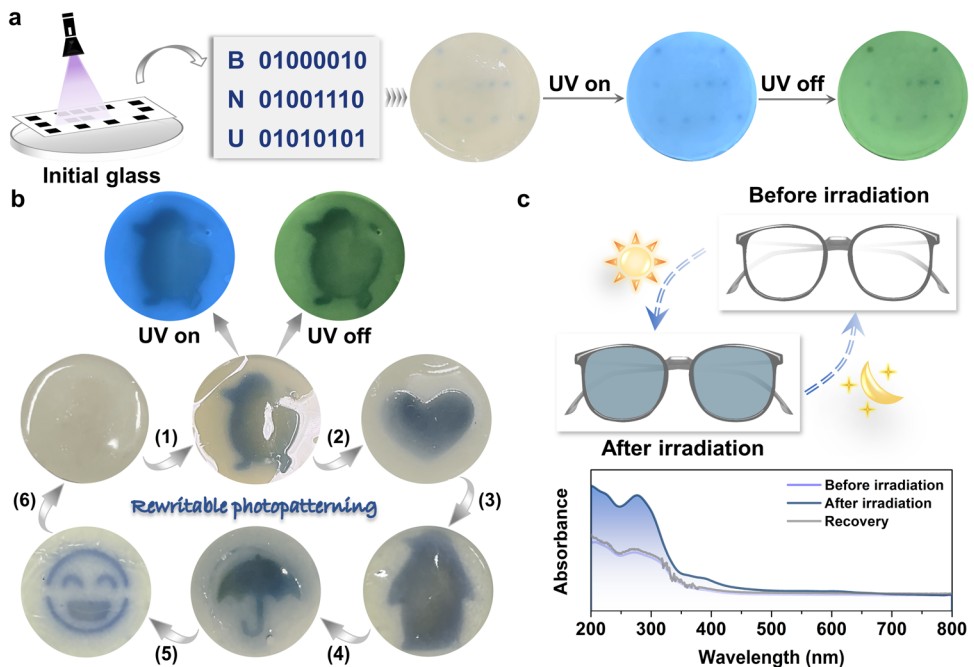

**Fig. 5 | Demonstration of reversible photochromic ultralong phosphorescence emissions of the glass for potential applications. a** The "BNU" alphabet was recorded onto the transparent glass using a binary format. The photographs were captured using an iPhone. **b** Writing and recovering photochromic patterns through alternating 365 nm light irradiation and heating. **c** Photochromic behavior under sunlight irradiation, along with absorption spectra of sunglasses before and after sun exposure and recovery. The sunglass model images were created using Microsoft PowerPoint.

demonstrates the realization of circular polarized phosphorescence responses in these OIMHs glasses. The magnitude of the circular polarization at the excited state can be evaluated by the optical dissymmetry factor ($g_{lum} = 2(I_L - I_R)/(I_L + I_R)$), where $I_L$ and $I_R$ are respectively the emission intensities of left and right CPL[48]. The $g_{lum}$ values for P⁺-based glasses (P-Zn, P-Zn-Sb, and P-Sb glasses) reach the order of $10^{-2}$ (Supplementary Fig. 38), which are at least comparable to or even higher than those observed in many amorphous chiral systems[49–54]. These exceptional chiro-optical properties are speculated to originate from the broken mirror symmetry upon aggregation of P⁺-based OIMHs. Furthermore, the well-ordered packing and restricted non-radiation transitions in the confined environment may contribute to the large emission dissymmetry of the CPL-active glasses[48]. With their distinctive CPL characteristics, these transparent glasses hold substantial potential for the development of chiral photoelectric devices, particularly in the domains of optical displays and information storage.

## Application of the dynamic RTP glass

The escalating demand for high-density storage media has surged alongside the rapid expansion of digital information[55,56]. Enhancing storage capacity involves integrating multiple dimensions—wavelength, time, space, and luminescence—into a unified optical carrier[57]. In contrast to prevalent 2D storage media[58–60], the large transparent P-Zn-BP glass, characterized by its distinct three-dimensionality, holds the potential for achieving heightened optical storage capacity. Leveraging the photo-regulated ultralong RTP and photochromic characteristics of the OIMHs glass, reversible 3D optical data storage was explored for practical applications. Under 365 nm UV illumination through printed maskers (Fig. 5a), binary dot arrays encoded by a standard computer file format were replicated into the raw glass (Supplementary Movie 1). Points undergoing color change and those remaining unchanged represented binary codes "1" and "0", respectively, generating binary data spelling out the "BNU" alphabet. In the absence of irradiation treatment, the fabricated pattern exhibited no discernible signal output under daylight. The color-unchanged section

emitted blue and green afterglow before and after photoirradiation in a dark environment, respectively. As such, diverse patterns of photochromism, fluorescence, and persistent luminescence in the bulk glass could be discerned, highlighting the potential of P-Zn-BP glass as a higher-density storage medium compared to many reported photochromic systems lacking long-lived RTP emission[61–64]. Simultaneously, these photochromic and RTP properties conferred a high-level encryption feature on the array. The pattern could be effortlessly erased by leaving the glass in the dark, offering possibilities for reversible high-density data storage and multiplexed coding capabilities. Rewritable photopatterning and anti-counterfeiting could similarly be accomplished using BP-doped glass, capitalizing on its reversible photo-modulated color change and tunable RTP characteristics (Fig. 5b).

Given that prolonged exposure to UV light from the sun may lead to a series of eye diseases, the development of UV-blocking, reusable and cost-effective sunglasses assumes great significance. The P-Zn-BP glass undergoes a gradual transition from colorless to blue upon exposure to sunlight for about 2 min, returning to its original state after being left in the dark (Fig. 5c). This observation prompts exploration of its suitability as photochromic sunglasses with high reversibility. Analysis of its absorption spectra confirms a noticeable increase in absorption intensity in the 200–400 nm range following sunlight exposure, with the initial absorption state fully recoverable. In essence, the reusable color-changing sunglasses could effectively shield against a wide range of sunlight wavelengths.

## Discussion

In conclusion, we present a type of transparent large-scale glasses, facilely fabricated through a convenient grinding (crystallization)-melting-quenching process. This method significantly expands the materials family of OIMHs beyond their typical forms of single crystals, powders, and thin films. Through the incorporation of diverse ions and organic units, we achieve a wide range of luminescence, ranging from visible to NIR, with tunable lifetimes spanning from microseconds to

seconds. Notably, the introduction of BP into P-Zn glass enables dynamic ultralong RTP and a photochromic switchable functionality. The emergence of prolonged RTP is highly linked to the robust hydrogen-bonding network, heavy atom effects, and the high hardness and density in the rigid glassy structure. The persistent luminescence is easily regulated by photo-generated radicals. In the absence of chiral molecules, the prepared glasses exhibit distinctive chiro-optical properties, arising from the broken mirror symmetry upon aggregation of OIMHs containing propeller-like configured organic units. Harnessing the ultralong RTP and photochromism, we achieve noteworthy demonstrations, including 3D optical data storage, multiple encryptions, and rewritable information recording. Furthermore, we identify the potential of P-Zn-BP glass as photochromic sunglasses. Hence, leveraging inexpensive raw materials and facile preparation, this work not only presents an alternative route for synthesizing intelligent photo-responsive bulk glasses with tunable RTP and reversible photochromism, but also opens avenues for applying these OIMHs glasses as a versatile platform in diverse advanced photo-functional and photonic applications.

## Methods

### Materials

(methoxymethyl)triphenylphosphonium chloride (P-Cl) (Adamas, 99%+), (methoxymethyl)triphenylphosphonium bromide (P-Br) (Maya, 99% + ), 4,4´-bipyridine (BP) (Psaitong, 99%), zinc chloride ($ZnCl_2$) (Macklin, 99%), zinc bromide ($ZnBr_2$) (Innochem, 99.9%), zinc iodide ($ZnI_2$) (Innochem, 99%), antimony trichloride ($SbCl_3$) (Innochem, 99.9%), polyvinyl alcohol (PVA) (Innochem, MW ≈ 20,000), polyvinylpyrrolidone (PVP) (Innochem, K30) and gelatin (GEL) (Macklin, 99%) were purchased as indicated and used without further purification. Ethanol (Beijing Oriental Shibo Fine Chemical Co., LTD, ≥99.5%) was purchased as indicated and used without further purification. Deionized water was utilized throughout the whole experimental process.

### Synthesis

**Preparation of P-Zn crystal.** P-Cl powder and $ZnCl_2$ (molar ratio: 2:1) were separately dissolved in the deionized water and ethanol with the aid of sonication to form clear solutions. Then, the two solutions were mixed in an open glass bottle and evaporated naturally at room temperature. After some days, the blocky transparent crystals were obtained.

**Preparation of P-Zn glass.** The as-prepared P-Zn crystal or the uniformly ground mixture of P-Cl and $ZnCl_2$ in a 2:1 molar ratio was transferred to a silicone mold. The mold was then placed in a preheated oven set at 140 °C. A transparent melt was formed within the mold after heating for 40 min at 140 °C, and the mold with melt inside was shaken to remove the bubbles. Following this, the mold was taken out and allowed to cool naturally to room temperature. The homogeneous glass formed within 3 min.

**Preparation of P-Zn-Br/P-Zn-I/P-Zn-Sb/P-Sb glasses.** These glasses could be prepared using the same procedure as P-Zn glass, with variations in the starting materials. Specifically, for P-Zn-Br glass, raw materials of P-Cl and $ZnBr_2$ were used in an exact 2:1 molar ratio. For P-Zn-I glass, raw materials of P-Cl and $ZnI_2$ were used in an exact 2:1 molar ratio. For P-Zn-Sb glass, raw materials of P-Cl, $ZnCl_2$, and $SbCl_3$ were used in an exact molar ratio of 2:0.9:0.1. In the case of P-Sb glass, raw materials of P-Cl and $SbCl_3$ were used in an exact 2:1 molar ratio.

**Preparation of BP-doped P-Zn-BP/PBr-Zn-Br-BP glasses.** These glasses could be prepared using the same procedure as P-Zn glass, with variations in the starting materials. For the P-Zn-BP glass, raw materials of P-Cl, $ZnCl_2$, and BP were used in a molar ratio of 2:1:0.02. For PBr-Zn-

Br-BP glass, the raw materials of P-Br, $ZnBr_2$, and BP were used in an exact molar ratio of 2:1:0.02.

**Preparation of BP-doped polymeric PVA-BP, PVP-BP, and GEL-BP films.** PVA/PVP/GEL and BP (mass ratio: 1:0.02) were each dissolved in water and ethanol to form clear solutions, respectively. Subsequently, the BP alcoholic solution was added into the PVA/PVP/GEL aqueous solution under stirring. The mixed solutions were then placed in an oven and heated at 40 °C to remove solvents, and corresponding BP-doped films could be obtained after 24 h.

### Characterizations

**X-ray crystallography.** The single-crystal X-ray diffraction data were collected on a Rigaku XtalLAB Synergy diffractometer with Cu-Kα radiation ($\lambda = 1.54184$ Å). Olex2 software was used to solve and refine the structure.

### Nanoindentation measurement

The Young's modulus of the P-Zn glass was tested by using the NanoTest Vantage (MML, UK) equipped with a three-sided pyramidal (Berkovich) diamond indenter tip (Young's modulus: 1141 GPa; Poisson's ratio: 0.07). Nanoindentation measurements were performed at a constant strain rate of 0.05 s⁻¹. The hold time was 20 s and the depth limit was 500 nm. Four points of the sample were randomly selected for testing.

### Solid-state $^{13}C$ NMR spectra

Magic angle spinning solid-state $^{13}C$ NMR spectra were recorded at 100.64 MHz with high-power proton decoupling on the Bruker Avance III 400 MHz spectrometer equipped with MAS probe for a 2.5 AND 4-mm rotor, with a relaxation delay of 2 s, a recycle delay of 4 s. The spinning frequencies of the spinner were stable within ±5 Hz. $^{13}C$ chemical shifts were measured indirectly by reference to the carbonyl α-glycine line set at 176.5 ppm. The spectral data were analyzed by using MestReNova software.

### FT-IR measurement

FT-IR spectra were performed on a Bruker TENSOR 27 infrared spectrophotometer. The as-prepared samples were mixed with potassium bromide (KBr) and pressed as pellets before measurement. The spectra were recorded by performing 32 scans between 4000 and 400 cm⁻¹.

### HR-ESI-MS measurement

HR-ESI-MS spectra were measured on a quadrupole time-of-flight (Q-TOF) mass spectrometer (Q-TOF liquid chromatography/mass spectrometry (LC/MS) 6540 series, Agilent Technologies, Santa Clara, CA) coupled with electrospray ionization (ESI). The mixed solvents of deionized water and ethanol were used for the sample measurements.

### Thermal analysis

DSC measurements were performed on the METTLER TOLEDO DSC 1 calorimeter under $N_2$ atmosphere with a heating rate of 5 °C min⁻¹. The TGA analyses were carried out on NETZSCH STA449 F5 Thermogravimetric Analyzer with a heating rate of 10 °C min⁻¹.

### PXRD measurement

PXRD data were performed on a Rigaku Ultima-IV automated diffraction system with Cu Kα radiation ($\lambda = 1.5406$ Å), and the operating power was 40 kV, 30 mA. The measurements were made in a 2θ range of 5°–50° with a step of 0.02° (2θ) as well as a scan speed of 5° min⁻¹. The simulated PXRD curve for P-Zn crystal was generated using the single-crystal data and diffraction-crystal module of the Mercury (Hg) program.

## UV-Vis-NIR transmittance/absorption measurement

The transmittance/absorption spectra of the samples were carried out on a SPECORD200 spectrophotometer.

## Fluorescence microscope observations

The fluorescence microscopy images were obtained by using fluorescence microscope (OLYMPUS IXTI) equipped with a multispectral imaging system.

## SEM measurement

SEM images were obtained using a Hitachi SU-8010 instrument. To decrease charging effects, the samples were sputtered with gold prior to the measurement.

## TEM measurement

TEM images and SAED patterns were performed on Talos F200S operated at 200 kV.

## Density measurement

The density tests were performed on a density measurement device (America-Micromeritics, AccuPyc-1330/1340) by using helium gas replacement. Each sample underwent five separate tests, and the reported results represented the average values.

## PL measurement

All the relevant PL tests were recorded on an Edinburgh FLS-980 fluorescence spectrometer with a xenon arc lamp (Xe900) and a microsecond flash-lamp (uF900). The fluorescence decay profiles of the samples were obtained by using a picosecond pulsed diode laser. The long-lived phosphorescence lifetime ($\tau_p$) was evaluated by individual component lifetimes $\tau_i$ and amplitudes $A_i$ of $\tau_i$ in double- or triple-exponential decay profiles. For a double-exponential decay, the lifetime was calculated using the equation: $\tau_p = (A_1\tau_1^2 + A_2\tau_2^2)/(A_1\tau_1 + A_2\tau_2)$. Similarly, in the three-exponential case, $\tau_p = (A_1\tau_1^2 + A_2\tau_2^2 + A_3\tau_3^2)/(A_1\tau_1 + A_2\tau_2 + A_3\tau_3)$. The fitting goodness was evaluated by the value of $\chi^2$, which ideally should be lower than 1.300.

## EPR measurement

EPR signals were recorded with a Bruker A300 system (modulation frequency: 100.00 KHz, modulation amplitude: 2.00 G, sweep width: 100.00 G, time constant: 40.960 ms, sweep time: 60.7 s, microwave power: 20.00 mW, frequency: 9.84 GHz). The energy difference ($\Delta E$) studied in EPR spectroscopy mainly arises from the interaction of unpaired electrons within the sample with a magnetic field ($B_0$), which is quantified by the equation: $\Delta E = g_e\beta B_0$, where $\beta$ represents Bohr magneton and $g_e$ is the spectroscopic g-factor of the free electron and equals 2.0023192778. However, due to the dependence of spin-orbit coupling (SOC), the calculation of energy difference is further modified to: $\Delta E = g\beta B_0$. Organic free radicals with only C, H, O, and N atoms, produce $g$ factors very close to $g_e$, because of the small contribution from SOC, while the $g$ factors of much larger elements (such as metals) may be significantly different from $g_e$[65].

## CD and CPL signals measurement

CD spectra of the samples were obtained on a JASCO J-1500 CD spectrometer. The CPL spectra of the glassy samples were recorded on JASCO CPL-200 spectrometer using an external excitation source, 320 nm semiconductor laser. Before measurement of the CPL spectra, the glassy film was fixed to the sample holder, allowing the excited light source to pass through the film with a thickness not exceeding 2 mm.

## Theoretical calculations

Density functional theory (DFT) calculations were carried out using Gaussian 16 programs throughout this manuscript[66]. Geometric optimizations were performed using B3LYP functional[67] with Grimme's dispersion correction of D3 version (Becke-Johnson damping). The standard 6-311G** basis set[68] was used for H, C, N, O, P, and Cl, while SDD basis set and corresponding effective core potential were used for Zn. Time-dependent density functional theory (TDDFT) calculations were performed on the optimized structures at the same theoretical level. The first 10 $S_0 \rightarrow S_n$ vertical transitions were calculated. The analyses of electrostatic potential (ESP) on molecular van der Waals (VDW) surface and hole-electron distribution were finished by Multiwfn[69]. The above isosurfaces were rendered by VMD program based on the outputs of Multiwfn.

## Reporting summary

Further information on research design is available in the Nature Portfolio Reporting Summary linked to this article.

## Data availability

The data that support the findings of this study have been included in the main text and Supplementary Information. All other information can be obtained from the corresponding author upon request. Crystallographic data generated in this study have been submitted to the Cambridge Crystallographic Data Centre (www.ccdc.cam.ac.uk/data_request/cif) as supplementary publication no. CCDC: 2323188.

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

## Acknowledgements

This study was supported by the National Natural Science Foundation of China (Grant No. 22275021), the Beijing Municipal Natural Science Foundation (Grant No. L234064), the Beijing Nova Program (Grant No. 20230484414), the Shandong Laboratory of Advanced Materials and Green Manufacturing at Yantai (AMGM2024F23), and the Fundamental Research Funds for the Central Universities.

## Author contributions

D.Y. and F.N. conceived the experiments. F.N. conducted and analyzed the experiments. D.Y. supervised the project. The two authors prepared and edited the manuscript.

## Competing interests

The authors declare no competing interests.
