## [Peer Review File · Nature Communications]

Zero-dimensional Halide Hybrid Bulk Glass Exhibiting Reversible Photochromic Ultralong PhosphorescenceReviewer #1 (Remarks to the Author):

In this manuscript, Nie and Yan present a facile and efficient strategy to design macroscopic hybrid glass with reversible photochromic ultralong phosphorescence. The combined grinding-melting-quenching process is highly useful and convenient. Interestingly, the incorporation of the aggregation-induced chirality effect leads to intriguing chiro-optical properties of the glass. Moreover, this work demonstrates promising applications, including 3D optical storage, rewritable photo-patterning and multi-mode anti-counterfeiting with ease. Overall, this is a highly compelling work, and I believe this manuscript is worth of publication on Nature Communications following some minor revision. Details need to be further clarified as shown below:

1. The authors may discuss the requirements or give the standards for selecting organic-inorganic units in the preparation of organic-inorganic halides glasses via the melting-quenching method.
2. g value stands for a crucial parameter for CPL emission. The authors mentioned that the obtained g values are at least comparable to or even higher than those observed in many amorphous chiral systems. So, what is the decisive parameter for controlling g ?
3. The authors pointed out that limited efforts have been directed towards the fabrication of glassy OIMHs due to their relatively weak glass formation ability. Why do organic-inorganic metal halides exhibit weak glass formation ability?
4. For the ultralong phosphorescence lifetime values, how did the authors obtain the fitting results and fitting goodness based on the decay curves?
5. Some figures (such as Figure 3 (a), Figure 5 (a) (b), Supplementary Figure 25, and Supplementary Figure 31) in the Manuscript and Supplementary Information lack scale bars.

Reviewer #2 (Remarks to the Author):

The manuscript entitled "OD Halide Hybrid Bulk Glass Exhibiting Reversible Photochromic Ultralong Phosphorescence" presents a meaningful investigation of organic-inorganic hybrid halide glasses that were doped with variable photoactive units (including both luminescent metal ions and photochromic 4,4'-bipyridine). The authors have conducted meticulous experimental studies (such as amorphous structure, thermal analysis and phosphorescence), along with theoretical investigations on the inorganic-organic hybrid glasses. Also, the extensive appendix largely supports the understanding and interpretation of the new type of glasses with both reversible color-changing and ultralong phosphorescence. In all, this is an interesting and sound work, and I suggest that this work could be published in Nature Communications after addressing some minor concerns:

1. The authors mentioned "the EPR spectrum shows a conspicuous signal characterized by a g value of 2.0016 after irradiation, while no signal is discernible prior to exposure." What's the meaning of " g " here? What range of g value could indicate the generation of radicals?
2. The authors should clarify how circularly polarized luminescence (CPL) signals were tested for glassy samples in this work.
3. Why did only one peak appear in the prompt spectra of P-Zn-BP glass (as shown in Figure 3c)?
4. The authors show that the BP doping into traditional polymers (such as polyvinyl alcohol) has no photochromic character. They need to give more comments about the reason for this point.
5. The numbers in Supplementary Figure 1 are too small to be clearly discernible. Please enlarge them for better visibility.
6. The author claims that the blue emission of P-Zn-BP glass before UV excitation originates from P^+ cations, while the emission after coloration seems to come from either radiative transition of BP anionic radicals or radiative transition of P^+ cations with the energy level of BP radicals as mediator. Any mechanism is not completely consistent with the mechanism shown in Figure S28.
7. Please explain why BP radicals can be generated through charge transfer in various light sources including X-ray, ultraviolet, and visible light. Perhaps UV-vis absorption spectra of P-Zn-BP model in Figure S29 by theoretical calculation may provide some assistance.

Reviewer #3 (Remarks to the Author):

The authors presented a work on the fabrication of bulk hybrid glassy OIMHs that demonstrate simultaneous photochromism and dynamic photo-responsive RTP for the first time. Though it is unclear if the combination of these two optical behaviours will have a near term advantage in any applications, the possibility to combine these two behaviours is interesting. Due to the organization of P⁺ in the P-Zn-BP system, chiral behaviour is also present in the system. Here are some questions that I have:

A "glassy OIMHs" this term is challenging to understand. OIMH is crystalline... so this is literally Glassy Crystalline Organic-Inorganic MHs. Is the system more like nanocrystalline particles/clusters dispersed in the glassy substrate, in this case glassy ZnP? Any TEM image of a thin section of this glass?

The author mentioned "The significantly prolonged RTP lifetime in P-Zn glass is attributed to strong hydrogen bonding interactions between organic cations (P⁺) and inorganic anions (ZnCl₄²⁻), external heavy atom effects of Zn and Cl atoms, and the rigid glassy structure that facilitates the intersystem crossing process." What do you mean? Do you mean the phosphorescence is due to spin-orbit coupling? Any proof?

What is the reason for the photochromic behaviour? Is this a result of ring closure? This is unclear.

The author says that the reversible RTP switching is demonstrated to be repeatable for at least four cycles. What happens after the 4th cycles?

From Figure 3e, there seems to be multiple decay rates. What are they associated with?

Reviewer #4 (Remarks to the Author):

This is an interesting work reporting the synthesis and investigation of a new photochromic glass. The work appears as well conducted and the manuscript is clear and well written. Overall it appears as a high quality work reporting some new interesting insights and materials that may have some interesting applications. I do miss however, the comparison of the merits of this new system with state of the art photochromic materials. This comparison could also be made as a form of table that will surely enrich the presentations of the work.

Reviewer comments:

Reviewer #1:

In this manuscript, Nie and Yan present a facile and efficient strategy to design macroscopic hybrid glass with reversible photochromic ultralong phosphorescence. The combined grinding-melting-quenching process is highly useful and convenient. Interestingly, the incorporation of the aggregation-induced chirality effect leads to intriguing chiro-optical properties of the glass. Moreover, this work demonstrates promising applications, including 3D optical storage, rewritable photo-patterning and multi-mode anti-counterfeiting with ease. Overall, this is a highly compelling work, and I believe this manuscript is worth of publication on Nature Communications following some minor revision. Details need to be further clarified as shown below:

Response: We greatly appreciate the positive comments and very useful advice. Herein, we address the comments as follows:

1. The authors may discuss the requirements or give the standards for selecting organic-inorganic units in the preparation of organic-inorganic halides glasses via the melting-quenching method.

Response: Thank you very much for the insightful question. To achieve organic-inorganic metal halides (OIMHs) glasses through melting-quenching process, the first prerequisite is to access a stable liquid state of OIMHs. This requires that the melting temperature (T_m) is lower than the decomposition temperature (T_d), through either decreasing T_m or increasing T_d . OIMHs offer rich chemical tunability, allowing for materials with distinct thermal behaviors by selecting different organic components, metal ions and halide ions. Typically, the dissociation of organic components at high temperature triggers initial decomposition. Thus, selection of organic parts with less volatile cations forming stronger interactions with metal ions would stabilize OIMH structures and elevates T_d (such as *Angew. Chem. Int. Ed.*, 2023, 62, e202302406). Additionally, the quenching process should also be noted, as some liquid-state OIMHs tend to crystallize under cooling conditions. A promising approach involves employing organic cations with relatively large molecular size and structural complexity to prevent

crystallization of OIMHs during cooling, which facilitates the glass formation (such as *Chem. Rev.*, 2022, 122, 4163).

In this case, the formation of a series of new OIMHs glasses can be attributed to the meticulous selection of organic constituents, ensuring relatively high T_d and low T_m of the OIMHs. Specifically, the high stability (T_d , approximately 300 °C) of organic P-Cl contributes to the elevated T_d of the OIMHs (Supplementary Fig. 9b). Moreover, the incorporation of P^+ cations with large molecular sizes and tetrahedral clusters of $ZnCl_4^{2-}$ units may prevent ordering during the quenching process (such as *Chem. Rev.*, 2022, 122, 4163).

One paragraph has been updated in Page 8, line 148, Revised Manuscript: In this case, the formation of a series of new OIMHs glasses can be attributed to the meticulous selection of organic constituents, ensuring relatively high T_d and low T_m of the OIMHs. Specifically, the high stability (T_d , approximately 300 °C) of organic P^+ cations contributes to the elevated T_d of the OIMHs (Supplementary Fig. 9b). Moreover, the incorporation of P^+ cations with large molecular sizes and tetrahedral clusters of $ZnCl_4^{2-}$ units may prevent ordering during the quenching process²⁷.

The corresponding reference has been added in the Revised Manuscript as below:

27. Ma, N. and Horike, S. Metal–Organic Network-Forming Glasses. *Chem. Rev.* **122**, 4163–4203 (2022).

2. g_{lum} value stands for a crucial parameter for CPL emission. The authors mentioned that the obtained g_{lum} values are at least comparable to or even higher than those observed in many amorphous chiral systems. So, what is the decisive parameter for controlling g_{lum} ?

Response: Thank you very much for the professional comment. For developing CPL materials, a critical concern is to achieve a high luminescence dissymmetry factor (g_{lum}), which serves as a metric for quantifying the level of CPL. In CPL spectroscopy, the differential emission intensity (ΔI) between the left- (I_L) and right-handed polarized luminescence (I_R) can be measured upon exposure to unpolarized excitation light. The g_{lum} , as a function of wavelength, is defined as the relative difference between I_L and

I_R , expressed as $g_{lum} = 2 \times (I_L - I_R)/(I_L + I_R)$. Theoretically, g_{lum} can be calculated using the equation: $g_{lum} = 4(|\mu||m|\cos\theta_{\mu,m})/(|\mu|^2 + |m|^2)$ (such as *Angew. Chem. Int. Ed.*, 2022, 61, e202204609; *Chem. Rev.*, 2021, 121, 2373), where u and m represent the electric and magnetic transition dipole moments, respectively, and $\theta_{\mu,m}$ represents the angle between these two dipole moments. $|g_{lum}|$ reaches its maximum value of 2 when u and m are equal to each other in length and orientation in either the parallel or antiparallel direction. However, given that the length of m is typically much smaller than that of μ in most small organic molecules, the denominator in the equation is primarily influenced by $|\mu|^2$, leading to the simplified equation: $g_{lum} \approx 4|m|\cos\theta_{\mu,m}/|\mu|$. Consequently, a large dissymmetry factor can be achieved through electrically forbidden transitions (i.e., small $|\mu|$) and magnetically allowed transitions (such as *Chem. Rev.*, 2021, 121, 2373). Due to the large $|\mu|$ and negligible $|m|$ in chiral organic emitters, achieving highly efficient CPL activity becomes challenging.

Self-assembly has been demonstrated as an efficient method for amplifying the g_{lum} of many CPL-active materials (such as *Adv. Mater.*, 2020, 32, 1900110). In chiral assemblies, the enhanced packing order of the molecules and the significantly constrained non-radiative transitions within the confined environment, contribute to a greater emission dissymmetry (such as *Chem. Rev.*, 2015, 115, 7304). Through self-assembly, in addition to chiral molecules, completely achiral molecules can form chiral supramolecular assemblies. This phenomenon is related to spontaneous symmetry breaking (such as *Chem. Rev.*, 2015, 115, 7304; *Chem. Rev.*, 2021, 121, 2147). In this study, the chirality of P-Zn OIMH, initially an achiral molecule, is induced when its three phenyl rings are fixed in a preferred clockwise or anticlockwise orientation in the aggregated state. We infer that the high g_{lum} observed in the P⁺-based glasses stems from the well-ordered aggregation and reduced non-radiation transition in the assemblies (such as *Chem. Rev.*, 2015, 115, 7304).

One paragraph has been updated into Page 16, line 310, Revised Manuscript: These exceptional chiro-optical properties are speculated to originate from the broken mirror symmetry upon aggregation of P⁺-based OIMHs. Furthermore, the well-ordered

packing and restricted non-radiation transitions in the confined environment may contribute to the large emission dissymmetry of the CPL-active glasses⁴⁸.

The corresponding reference has been added in the Revised Manuscript as below:

48. Liu, M., Zhang, L. and Wang, T. Supramolecular Chirality in Self-Assembled Systems. *Chem. Rev.* **115**, 7304–7397 (2015).

3. The authors pointed out that limited efforts have been directed towards the fabrication of glassy OIMHs due to their relatively weak glass formation ability. Why do organic-inorganic metal halides exhibit weak glass formation ability?

Response: We thank the reviewer's valuable question. The mainstream method of preparing OIMHs glasses is the melt-quenched method, as demonstrated in several recent studies (such as *Angew. Chem. Int. Ed.*, 2023, 62, e202302406; *J. Am. Chem. Soc.*, 2024, 10.1021/jacs.3c12296). However, a critical challenge in fabricating glassy OIMHs for this approach is that most OIMHs exhibit a decomposition temperature lower than their melting point. Additionally, the quenching process may lead to unavoidable recrystallization due to the low formation energy of metal halides (such as *Adv. Mater.*, 2021, 33, 2005868; *Sci. Adv.*, 2018, 4, eaao6827). Thus, employing bulky organic cations with high spatial resistance to mitigate crystallization tendency may offer a promising strategy for producing OIMHs glass (such as *Nat. Chem.*, 2021, 13, 778). Consequently, the fabrication of glasses from OIMHs remains challenging, and the field of such hybrid glasses is still in its nascent stage.

One paragraph has been updated into Page 3, line 60, Revised Manuscript: However, limited efforts have been directed towards the fabrication of glassy OIMHs due to their relatively weak glass formation ability. This is attributed to the facile dissociation of organic components prior to the melting of OIMHs using the mainstream melt-quenched approach, alongside a pronounced tendency for crystallization upon cooling²¹.

The corresponding reference has been added in the Revised Manuscript as below:

21. Ye, C., McHugh, L. N., Chen, C., Dutton, S. E. and Bennett, T. D. Glass Formation in Hybrid Organic-Inorganic Perovskites. *Angew. Chem. Int. Ed.* **62**, e202302406 (2023).

4. For the ultralong phosphorescence lifetime values, how did the authors obtain the fitting results and fitting goodness based on the decay curves?

Response: Thank you very much for the valuable comment. The long-lived phosphorescence lifetime (τ_p) is evaluated by individual component lifetimes τ_i and amplitudes A_i of τ_i in double- or triple-exponential decay profiles (such as *Angew. Chem. Int. Ed.*, 2020, 59, 23067). For a double-exponential decay, the lifetime is calculated using the equation: $\tau_p = (A_1\tau_1^2 + A_2\tau_2^2)/(A_1\tau_1 + A_2\tau_2)$. Similarly, in the three-exponential case, $\tau_p = (A_1\tau_1^2 + A_2\tau_2^2 + A_3\tau_3^2)/(A_1\tau_1 + A_2\tau_2 + A_3\tau_3)$. The fitting goodness is evaluated by the value of χ^2 , which ideally should be lower than 1.300.

One paragraph has been added into Page 3, line 78, Revised Supplementary Information: The long-lived phosphorescence lifetime (τ_p) was evaluated by individual component lifetimes τ_i and amplitudes A_i of τ_i in double- or triple-exponential decay profiles. For a double-exponential decay, the lifetime was calculated using the equation: $\tau_p = (A_1\tau_1^2 + A_2\tau_2^2)/(A_1\tau_1 + A_2\tau_2)$. Similarly, in the three-exponential case, $\tau_p = (A_1\tau_1^2 + A_2\tau_2^2 + A_3\tau_3^2)/(A_1\tau_1 + A_2\tau_2 + A_3\tau_3)$. The fitting goodness was evaluated by the value of χ^2 , which ideally should be lower than 1.300.

5. Some figures (such as Figure 3 (a), Figure 5 (a) (b), Supplementary Figure 25, and Supplementary Figure 31) in the Manuscript and Supplementary Information lack scale bars.

Response: The careful review from the referee is highly appreciated. The relevant scale bars have been added in the Revised Manuscript and Supplementary Information, as shown in Fig. R1–R5 below.

Fig. R1. UV-vis spectra of P-Zn-BP glass at irradiation times of 0, 10, 20, 40, and 60 s.

Inset: photographs of P-Zn-BP glass before and after 365 nm UV light irradiations.

Fig. R1 has been updated as Fig. 3a in the Revised Manuscript.

Fig. R2. The “BNU” alphabet was recorded onto the transparent glass using a binary format.

Fig. R2 has been updated as Fig. 5a in the Revised Manuscript.

Fig. R3. Writing and recovering photochromic patterns through alternating 365 nm light irradiation and heating.

Fig. R3 has been updated as Fig. 5b in the Revised Manuscript.

Fig. R4. The photographs of P-Zn-BP glass after the irradiation by Xe lamp, X-rays and sunlight.

Fig. R4 has been updated as Supplementary Figure 26 in the Revised Supplementary Information.

Fig. R5. The photographs of the ground powder composed of P-Cl, $ZnCl_2$ and BP (in a molar ratio of 2:1:0.02), BP-dope glasses (P-Sb-BP and PBr-Zn-Br-BP) and BP-doped polymeric films (PVA-BP, PVP-BP and GEL-BP) captured before and after exposure to a 365 nm lamp (5 W) for 5 min.

Fig. R5 has been updated as Supplementary Figure 32 in the Revised Supplementary Information.

Reviewer #2:

The manuscript entitled “0D Halide Hybrid Bulk Glass Exhibiting Reversible Photochromic Ultralong Phosphorescence” presents a meaningful investigation of organic-inorganic hybrid halide glasses that were doped with variable photoactive units (including both luminescent metal ions and photochromic 4,4'-bipyridine). The authors have conducted meticulous experimental studies (such as amorphous structure, thermal analysis and phosphorescence), along with theoretical

investigations on the inorganic-organic hybrid glasses. Also, the extensive appendix largely supports the understanding and interpretation of the new type of glasses with both reversible color-changing and ultralong phosphorescence. In all, this is an interesting and sound work, and I suggest that this work could be published in Nature Communications after addressing some minor concerns:

Response: Thank you for the kind recommendation of this work. We have carefully revised the manuscript according to the valuable suggestion.

1. The authors mentioned “the EPR spectrum shows a conspicuous signal characterized by a g value of 2.0016 after irradiation, while no signal is discernible prior to exposure.” What’s the meaning of “g” here? What range of g value could indicate the generation of radicals?

Response: We thank the reviewer very much for this valuable question. EPR spectroscopy, akin to many other techniques that depend on the absorption of electromagnetic radiation, serves as a valuable tool for elucidating the electronic structure of paramagnetic species (such as *Appl. Magn. Reson.*, 1992, 3, 219). Within a molecule or atom, discrete states exist, each corresponding to a specific energy level. Spectroscopy is the measurement and interpretation of the energy differences between the molecular or atomic states. According to Planck's law, the energy difference, denoted as ΔE , can be calculated *via* the equation: $\Delta E = h\nu$, where h represents Planck's constant and ν is the frequency of the radiation.

The absorption of energy triggers a transition from a lower energy state to a higher energy state. For EPR spectroscopy, the energy differences primarily stem from the interaction between unpaired electrons within the sample and the magnetic field generated by a magnet in the laboratory—an effect known as the Zeeman Effect (such as *Chem. Soc. Rev.*, 2018, 47, 2534). The magnetic field, B_0 , gives rise to two energy levels for the magnetic moment (μ) of the electron. The two states are labeled by the projection of the electron spin ($m_s, \pm 1/2$) on the direction of the magnetic field. For an electron, the magnetic moment is defined as: $\mu = m_s g_e \beta$, where β is a conversion constant called the Bohr magneton and g_e is the spectroscopic g-factor of the free electron and equals 2.0023192778. Consequently, the energies for an electron with

energy orientations of μ can be respectively defined as: $E_{1/2} = \frac{1}{2} g_e \beta B_0$ and $E_{-1/2} = -\frac{1}{2} g_e \beta B_0$. As a result, ΔE can be defined as: $\Delta E = g_e \beta B_0$.

The interaction between the ground state and excited states admixes small amounts of orbital angular momentum to the ground state. It is conventionally assumed that the spin-orbit coupling (SOC) term is proportional to the intrinsic spin angular momentum (s). This means we can simply combine both terms by just changing the value of g_e to g . Consequently, the energy differences can be calculated using the equation: $\Delta E = g \beta B_0$.

It is well-known that the magnitude of the SOC contribution depends on the size of the nucleus containing the unpaired electron. As a result, organic free radicals, with only H, C, N and O atoms, will exhibit minimal influence from SOC. Thus, their g factors are very close to g_e (≈ 2.0) (such as *Chem. Soc. Rev.*, 2018, 47, 2534; *ACS Appl. Mater. Interfaces*, 2023, 15, 149; *Chem. Commun.*, 2021, 57, 3154). On the other hand, the g factors of much larger elements, such as metals, may be significantly different from g_e and their g values can be any real number (such as *J. Am. Chem. Soc.*, 2019, 141, 2421; *Inorg. Chem.*, 1998, 37, 5180).

In this study, the 4,4'-bipyridine comprising only C, H, O, and N atoms was chosen as the electron acceptor in photochromism. The analysis reveals a g -factor (2.0016) close to 2.0 in the EPR spectroscopy upon irradiation (Fig. 3b), suggesting successful electron transfer and the generation of 4,4'-bipyridine radicals (such as *Angew. Chem. Int. Ed.*, 2022, 61, e202114100; *Mater. Chem. Front.*, 2022, 6, 2709).

One paragraph has been added into Page 3, line 86, Revised Supplementary Information: The energy difference (ΔE) studied in EPR spectroscopy mainly arises from the interaction of unpaired electrons within the sample under a magnetic field (B_0), which is quantified by the equation: $\Delta E = g_e \beta B_0$, where β represents Bohr magneton and g_e is the spectroscopic g -factor of the free electron and equals 2.0023192778. However, due to the dependence of spin-orbit coupling (SOC), the calculation of energy difference is further modified to: $\Delta E = g \beta B_0$. Organic free radicals with only C, H, O, and N atoms, produce g factors very close to g_e , because of the small contribution from

SOC, while the g factors of much larger elements (such as metals) may be significantly different from g_e^1 .

The corresponding reference has been added in the Revised Supplementary Information as below:

1. Roessler, M. M. and Salvadori, E. Principles and applications of EPR spectroscopy in the chemical sciences. *Chem. Soc. Rev.* **47**, 2534–2553 (2018).

2. The authors should clarify how circularly polarized luminescence (CPL) signals were tested for glassy samples in this work.

Response: We thank the reviewer for raising this important question. The CPL spectra of the glassy samples were directly recorded using a JASCO CPL-200 spectrophotometer. Before measurement of the CPL signal, the glassy film was fixed to the sample holder, allowing the excited light source to pass through the film with a thickness not exceeding 2 mm.

One paragraph has been added into Page 4, line 95, Revised Supplementary Information: The CPL spectra of the glassy samples were recorded on JASCO CPL-200 spectrometer using an external excitation source, 320 nm semiconductor laser. Before measurement of the CPL spectra, the glassy film was fixed to the sample holder, allowing the excited light source to pass through the film with a thickness not exceeding 2 mm.

3. Why did only one peak appear in the prompt spectra of P-Zn-BP glass (as shown in Figure 3c)?

Response: Thanks for the professional question. The investigation into the predominant emission band at 467 nm in the prompt PL spectrum of P-Zn-BP glass prior to irradiation has been conducted. Time-resolved PL decay curves of the glass reveal a 2.0 ns lifetime at 350 nm (Fig. R6) and a longer lifetime of 107.9 ns at 525 nm (Fig. 3e). Additionally, temperature-dependent prompt and delayed PL spectra exhibit a systematic decrease in intensity with rising temperature from 77 to 297 K (Supplementary Fig. 28). Notably, a significant overlap is observed between the

emission bands in the prompt and delayed PL spectra (Fig. 3d). Importantly, two emission bands at 350 and 525 nm appear for the glass after coloration (Fig. R6), possibly due to the phosphorescence intensity of the glass decreasing faster than the fluorescence intensity after illumination. These findings collectively indicate that the emissive band in the prompt PL spectrum of colorless P-Zn-BP glass stems from a combination of fluorescence and phosphorescence, a phenomenon commonly observed in other RTP materials (such as *Nat. Mater.*, 2021, 20, 1539; *Adv. Mater.*, 2021, 33, 2007571; *Chem. Sci.*, 2022, 13, 7429).

Fig. R6. (a) Prompt PL spectra P-Zn-BP glass at different irradiation times. (b) The fluorescence decay profile of P-Zn-BP glass at 350 nm.

Fig. R6 has been added as Supplementary Figure 27 in the Revised Supplementary Information.

One paragraph has been added into Page 13, line 225, Revised Supplementary Information: The investigation into the predominant emission band at 467 nm in the prompt PL spectrum of P-Zn-BP glass before irradiation has been conducted (Supplementary Fig. 27a). Time-resolved PL decay curves of the glass reveal a 2.0 ns lifetime at 350 nm (Supplementary Fig. 27b) and a longer lifetime of 107.9 ms at 525 nm (Fig. 3e). Additionally, temperature-dependent prompt and delayed PL spectra exhibit a systematic decrease in intensity with rising temperature from 77 to 297 K (Supplementary Fig. 28). Notably, a significant overlap is observed between the emission bands in the prompt and delayed PL spectra (Fig. 3c, 3d). Importantly, two emission bands at 350 and 467 nm appear for the glass after coloration, possibly due to the phosphorescence intensity of the glass decreasing faster than the fluorescence intensity after illumination. These findings collectively indicate that the emissive band

in the prompt PL spectrum of colorless P-Zn-BP glass stems from a combination of fluorescence and phosphorescence, a phenomenon commonly observed in many RTP materials^{13–15}.

The corresponding references have been added in the Revised Supplementary Information as below:

13. Ye, W. et al. Confining isolated chromophores for highly efficient blue phosphorescence. *Nat. Mater.* **20**, 1539–1544 (2021).
14. Zhou, B., Xiao, G. and Yan, D. Boosting Wide-Range Tunable Long-Afterglow in 1D Metal–Organic Halide Micro/Nanocrystals for Space/Time-Resolved Information Photonics. *Adv. Mater.* **33**, 2007571 (2021).
15. Zhou, B. and Yan, D. Color-tunable persistent luminescence in 1D zinc–organic halide microcrystals for single-component white light and temperature-gating optical waveguides. *Chem. Sci.* **13**, 7429–7436 (2022).

4. The authors show that the BP doping into traditional polymers (such as polyvinyl alcohol) has no photochromic character. They need to give more comments about the reason for this point.

Response: We thank the reviewer for raising this valuable question. For photochromic materials driven by electron transfer, their photo-responsive behavior is closely related to the electron-donating/accepting ability of the donor/acceptor. It has been confirmed that the photochromic performance of BP-doped glass is driven by reverse electron transfer between electron donor Cl anions and electron acceptor BP molecules. However, traditional polymers (such as polyvinyl alcohol, polyvinylpyrrolidone and gelatin) are mainly composed of C, H, O and N atoms, lacking suitable units for electron donation. Consequently, BP radicals cannot form in the BP-doped polymer films, resulting in the absence of photochromic property.

One paragraph has been added into Page 13, line 259, Revised Manuscript: Moreover, doping of BP into traditional polymer films such as polyvinyl alcohol, polyvinylpyrrolidone, and gelatin—comprising mainly C, H, O and N atoms—fails to

induce any photochromic behaviors. This deficiency may stem from the absence of suitable units for electron donation within these polymers.

5. The numbers in Supplementary Figure 1 are too small to be clearly discernible. Please enlarge them for better visibility.

Response: Thank you very much for the kind reminding. We have enlarged the numbers in Supplementary Figure 1 as below:

Fig. R7. The crystal structure of P-Zn as viewed from the a-axis (unit: Å).

Fig. R7 has been updated as Supplementary Figure 1 in the Revised Supplementary Information.

6. The author claims that the blue emission of P-Zn-BP glass before UV excitation originates from P⁺ cations, while the emission after coloration seems to come from either radiative transition of BP anionic radicals or radiative transition of P⁺ cations with the energy level of BP radicals as mediator. Any mechanism is not completely consistent with the mechanism shown in Figure S28.

Response: We thank the reviewer for providing us the opportunity to give more detailed explanation on luminescent behaviors of the BP-doped glass before and after coloration. The phosphorescence observed in BP-doped glass before irradiation mainly stems from the organic component, as evidenced by its RTP emission (Fig. 3d,

Supplementary Fig. 15a) and excitation spectra (Fig. R8) resembling those of P-Zn glass. Differently, the RTP emission observed in P-Zn-BP glass after coloration seems to arise from radiative transitions of P^+ cations with the energy level of BP radicals as a mediator (such as *Chem. Commun.*, 2021, 57, 3154; *Sci. China Chem.*, 2021, 64, 1297; *Angew. Chem. Int. Ed.*, 2023, 62, e202301564; *J. Am. Chem. Soc.*, 2022, 144, 6946). This is supported by the reduced intensity of phosphorescent emission (Fig. 3d) and excitation (Fig. R8), along with a slight blue-shift (from 523 to 514 nm) in the RTP emission following photoirradiation.

Fig. R8. Delayed PL excitation spectra of P-Zn glass and P-Zn-BP glass before and after photoirradiation.

Fig. R8 has been added as Supplementary Figure 29 in the Revised Supplementary Information.

One paragraph has been updated into Page 12, line 242, Revised Manuscript: As the photoirradiation time increases, the absorption intensity at 600 nm gradually decreases, indicating that heightened levels of self-absorption predominantly contribute to the decrease of RTP intensity and lifetime post-coloration (Fig. 3d, e). Moreover, there is a blue shift in RTP luminescence from 523 to 514 nm as the exposure time to light increases. These results indicate that alterations in chemical structure and absorbance induced by photo-irradiation facilitate the realization of the photo-stimuli reversible RTP (Supplementary Fig. 29)³⁵.

The corresponding reference has been added in the Revised Manuscript as below:

35. Ding, B., Gao, H., Wang, C. and Ma, X. Reversible room-temperature phosphorescence in response to light stimulation based on a photochromic copolymer. *Chem. Commun.* **57**, 3154–3157 (2021).

One paragraph has been added into Page 14, line 244, Revised Supplementary Information: The phosphorescence observed in BP-doped glass before irradiation mainly stems from the organic component, as evidenced by its RTP emission and excitation spectra resembling those of P-Zn glass (Fig. 3d, Supplementary Fig. 15a, 29). Differently, the RTP emission observed in P-Zn-BP glass after coloration seems to arise from radiative transitions of P^+ cations with the energy level of BP radicals as a mediator^{16, 17}. This is supported by the reduced intensity of phosphorescent emission and excitation, along with a slight blue-shift (from 523 to 514 nm) in the RTP emission following photoirradiation (Fig. 3d, Supplementary Fig. 29).

The corresponding references have been added in the Revised Supplementary Information as below:

16. Tao, Y. et al. Resonance-Induced Stimuli-Responsive Capacity Modulation of Organic Ultralong Room Temperature Phosphorescence. *J. Am. Chem. Soc.* **144**, 6946–6953 (2022).
17. Xu, Z. et al. Supercooled Liquids with Dynamic Room Temperature Phosphorescence Using Terminal Hydroxyl Engineering. *Angew. Chem. Int. Ed.* **62**, e202301564 (2023).

7. Please explain why BP radicals can be generated through charge transfer in various light sources including X-ray, ultraviolet, and visible light. Perhaps UV-vis absorption spectra of P-Zn-BP model in Figure S29 by theoretical calculation may provide some assistance.

Response: Thanks for your professional revision and valuable comments. As suggested by the reviewer, based on density functional theory calculations (Supplementary Fig. 30), distribution maps of hole and electron transitions from the ground state (S_0) to the singlet state (S_n) in the P-Zn-BP model reveal evident electron transfer between electron donor Cl anions and electron acceptor BP molecules. The electron transfer process

induces the formation of BP radicals, as observed in many other photochromic compounds based on electron transfer (such as *Coordin. Chem. Rev.*, 2022, 452, 214304; *Adv. Funct. Mater.*, 2023, 33, 2305796; *Adv. Funct. Mater.*, 2023, 33, 2212907; *Angew. Chem. Int. Ed.*, 2022, 61, e202114100).

One paragraph has been added into Page 13, line 249, Revised Manuscript: Based on density functional theory (DFT) calculations, distribution maps of hole and electron transitions from the ground state (S_0) to the singlet state (S_n) in the P-Zn-BP model reveal evident electron transfer between electron donor Cl anions and electron acceptor BP molecules (Supplementary Fig. 30, 31). The electron transfer process induces the formation of BP radicals, providing a competitive pathway for radiative transition in the form of phosphorescence.

Reviewer #3:

The authors presented a work on the fabrication of bulk hybrid glassy OIMHs that demonstrate simultaneous photochromism and dynamic photo-responsive RTP for the first time. Though it is unclear if the combination of these two optical behaviour will have a near term advantage in any applications, the possibility to combine these two behaviour is interesting. Due to the organization of P^+ in the P-Zn-BP system, chiral behaviour is also present in the system. Here are some questions that I have:

Response: Thanks for professional comments and kind recommendation of this work, especially the comment “the possibility to combine these two behaviour is interesting.” To date, molecule-based room temperature phosphorescence (RTP) with unique characteristics of long excited-state lifetimes, large Stokes shifts and high signal-to-noise ratios, has drawn considerable interest owing to the potential applications in encryption, anticounterfeiting, biomedicine, and so on (such as *Nat. Rev. Chem.*, 2023, 7, 800; *Nat. Rev. Chem.*, 2023, 7, 854; *Angew. Chem. Int. Ed.*, 2023, 62, e202302751). From a dynamic view, manipulating RTP on and off by external stimuli (including photo-stimulation, pressure, temperature, etc) is important to develop optical logical gates and smart luminescent switches (such as *Nat. Commun.*, 2024, 15, 2134; *Angew.*

Chem. Int. Ed., 2023, 62, e202314273). Particularly, the photo-responsive attributes, offering tunable emissive colors and lifetimes, present revolutionary potential in multilevel encryption and optical storage (such as *Angew. Chem. Int. Ed.*, 2020, 59, 4756; *Adv. Mater.*, 2021, 33, 2104002; *Mater. Horiz.*, 2023, 10, 5677). For instance, high-level information storage designs leveraging metal–organic frameworks crystals have demonstrated alterations in color, fluorescence, and RTP in response to irradiation stimuli (such as *Angew. Chem. Int. Ed.*, 2022, 61, e202114100). Exploiting the dynamic ultralong phosphorescence, this work has achieved promising applications, such as 3D optical storage, rewritable photo-patterning, and multi-mode anti-counterfeiting with ease. Therefore, this study introduces a smart hybrid glass platform as a new photo-responsive switchable system, offering versatility for a wide array of photonic applications.

One paragraph has been added into Page 2, line 30, Revised Manuscript:

Exploiting the dynamic ultralong phosphorescence, this work further achieves promising applications, such as 3D optical storage, rewritable photo-patterning, and multi-mode anti-counterfeiting with ease. Therefore, this study introduces a smart hybrid glass platform as a new photo-responsive switchable system, offering versatility for a wide array of photonic applications.

One paragraph has been added into Page 2, line 39, Revised Manuscript: Recently, there has been a noteworthy focus on dynamic room-temperature phosphorescence (RTP) featuring ultralong excited states^{4–6}. Particularly, the photo-responsive attributes, offering tunable emissive colors and lifetimes, present revolutionary potential in multilevel encryption and optical storage^{7–9}. For instance, information storage designs leveraging metal–organic frameworks crystals have demonstrated alterations in color, fluorescence, and RTP in response to irradiation stimuli¹⁰.

The corresponding references have been added in the Revised Supplementary Information as below:

4. Xu, Z. et al. Supercooled Liquids with Dynamic Room Temperature Phosphorescence Using Terminal Hydroxyl Engineering. *Angew. Chem. Int. Ed.* **62**, e202301564 (2023).

5. Tao, Y. et al. Resonance-Induced Stimuli-Responsive Capacity Modulation of Organic Ultralong Room Temperature Phosphorescence. *J. Am. Chem. Soc.* **144**, 6946–6953 (2022).
6. Li, J.-A. et al. Switchable and Highly Robust Ultralong Room-Temperature Phosphorescence from Polymer-Based Transparent Films with Three-Dimensional Covalent Networks for Erasable Light Printing. *Angew. Chem. Int. Ed.* **62**, e202217284 (2023).
7. Li, H. et al. Stimuli-Responsive Circularly Polarized Organic Ultralong Room Temperature Phosphorescence. *Angew. Chem. Int. Ed.* **59**, 4756–4762 (2020).
8. Yang, Y. et al. Tunable Photoresponsive Behaviors Based on Triphenylamine Derivatives: The Pivotal Role of π -Conjugated Structure and Corresponding Application. *Adv. Mater.* **33**, 2104002 (2021).
9. Song, T.-T. et al. Significant increase of the photoresponse range and conductivity for a chalcogenide semiconductor by viologen coating through charge transfer. *Mater. Horiz.* **10**, 5677–5683 (2023).
10. Ma, Y.-J., Fang, X., Xiao, G. and Yan, D. Dynamic Manipulating Space-Resolved Persistent Luminescence in Core–Shell MOFs Heterostructures via Reversible Photochromism. *Angew. Chem. Int. Ed.* **61**, e202114100 (2022).

The manuscript has been carefully revised according to your important suggestions.

1. A “glassy OIMHs” this term is challenging to understand. OIMH is crystalline... so this is literally Glassy Crystalline Organic-Inorganic MHs. Is the system more like nanocrystalline particles/clusters dispersed in the glassy substrate, in this case glassy ZnP? Any TEM image of a thin section of this glass?

Response: Thank you for the helpful comments. We appreciate the opportunity to provide further clarification on characteristic of glassy organic-inorganic metal halides (OIMHs). OIMHs represent a class of materials with the chemical composition of ABX, where A = organic cation, B = metal ion, and X = halogen ion (such as *Adv. Funct. Mater.*, 2023, 34, 2307896). The metal ion at the B-site is coordinated with the halogen ions at X-site, forming a structure wherein metal-halide polyhedra are enveloped by the

organic cations.

As noted by the reviewer, the research of OIMHs primarily targets crystalline materials (including single crystal, nanocrystal and polycrystalline film). However, the study rarely pays attention to the glassy counterparts. In recent years, an increased focus on the amorphization of OIMHs has been spurred by the growing interest in non-crystalline hybrid materials (such as *Nat. Chem.*, 2021, 13, 778). Melt-quenched glasses derived from OIMHs hold particular interest due to their demonstration of properties typically absent in their crystalline forms.

As we have known, crystal possesses a highly ordered and regular lattice structure, while the structure of glass lacks apparent periodicity in the arrangement of its molecules (such as *Chem. Rev.*, 2022, 122, 4163). In this study, both crystalline and glassy P-Zn share nearly identical chemical compositions but exhibit distinct microstructures. It is also noted that the glassy P-Zn does not behave as a system dispersed on a glassy substrate; rather, it can exist independently without reliance on any specific model.

Many experiments have been conducted to validate the aforementioned points. The structures of the targeted OIMHs crystal and glass are subsequently examined using solid-state ^{13}C nuclear magnetic resonance (NMR), high-resolution electrospray ionization mass (HR-ESI-MS) spectrometry, Fourier-transform infrared (FT-IR) spectroscopy, and/or X-ray diffraction (Fig. 2a–c, Supplementary Fig. 1, 2, 4, 7 and 8). These experiments verify that the glassy and crystalline samples have nearly identical organic-inorganic chemical compositions.

Significantly, the amorphous state of P-Zn can be achieved through heating the crystalline one, as assessed *via* temperature-dependent *in situ* powder X-ray diffraction (PXRD) analysis (Fig. 2b). For the crystalline sample, several diffraction peaks gradually vanish as the temperature rises from 25 to 100 °C. Upon reaching 140 °C, a broad “hump” band emerges in the PXRD pattern, signifying the formation of an amorphous liquid. This amorphous pattern persists after quenching to room temperature, indicating structural disordering in the glassy sample.

Based on your valuable suggestion, the high-resolution transmission electron microscopy (TEM) measurement has been further conducted (Fig. R9). The TEM images and the selected-area electron diffraction (SAED) patterns of P-Zn and BP-doped glasses reveal the absence of lattice fringes, further confirming their amorphous nature (such as *J. Am. Chem. Soc.*, 2016, 138, 10818; *Angew. Chem. Int. Ed.*, 2023, 62, e202218094).

Fig. R9. High-resolution TEM images and SAED patterns (inset) of (a) P-Zn and (b) P-Zn-BP glasses.

Fig. R9 has been added as Supplementary Figure 13 in the Revised Supplementary Information.

One sentence has been added into Page 8, line 160, Revised Manuscript: High-resolution transmission electron microscopy (TEM) images and the selected-area electron diffraction (SAED) patterns of P-Zn and BP-doped glasses show the absence of lattice fringes (Supplementary Fig. 13), further implying their amorphous nature^{28, 29}.

The corresponding references have been added in the Revised Manuscript as below:

28. Zhao, Y., Lee, S.-Y., Becknell, N., Yaghi, O. M. and Angell, C. A. Nanoporous Transparent MOF Glasses with Accessible Internal Surface. *J. Am. Chem. Soc.* **138**, 10818–10821 (2016).
29. Ali, M. A. et al. Fabrication of Super-Sized Metal Inorganic-Organic Hybrid Glass with Supramolecular Network via Crystallization-Suppressing Approach. *Angew. Chem. Int. Ed.* **62**, e202218094 (2023).

2. The author mentioned “The significantly prolonged RTP lifetime in P-Zn glass

is attributed to strong hydrogen bonding interactions between organic cations (P^+) and inorganic anions ($ZnCl_4^{2-}$), external heavy atom effects of Zn and Cl atoms, and the rigid glassy structure that facilitates the intersystem crossing process.”
What do you mean? Do you mean the phosphorescence is due to spin-orbit coupling? Any proof?

Response: We are grateful to the reviewer for granting us the opportunity to delve into the origins of phosphorescence in P-Zn glass. As commonly understood, the Jablonski diagram provides a schematic representation of various electronic states involved in luminescence phenomena (such as *Chem. Rev.*, 2017, 117, 6500). Following electronic absorption from S_0 to S_n and rapid internal conversion from S_n to S_1 in accordance with Kasha's rule, the singlet exciton is generated. Subsequently, it undergoes intersystem crossing (ISC) to generate triplet excitons. These triplet excitons then undergo several competing processes, including radiative decay to the ground state with phosphorescence emission, non-radiative decay to the ground state, reversible ISC from the excited triplet to singlet states, etc. Consequently, there are mainly two approaches to improving phosphorescence properties. One is promoting the ISC by introducing heavy atoms and/or organic moieties with lone-pair electrons. The heavy atoms are generally metallic ions and heavy halogen atoms, and possess strong spin-orbit coupling (SOC) capabilities (such as *J. Am. Chem. Soc.*, 2023, 145, 3937; *Coord. Chem. Rev.*, 2017, 346, 62). The other approach focuses on suppressing the quenching of triplet excitons by constraining non-radiative transitions through the construction of a rigid environment (such as *Nat. Rev. Chem.*, 2023, 7, 854; *Nat. Mater.*, 2021, 20, 1539).

Considering the above factors, herein, it is supposed that the notably prolonged RTP lifetime of P-Zn glass (124.0 ms) compared with that of P-Zn crystal (6.2 μ s) and P-Cl powder (3.3 μ s) is attributed to the heavy atom effects of Zn and Cl ions, which effectively facilitate the ISC process. Additionally, the increased density and stiffness of the glass matrix contribute to the formation of a more rigid structure, thereby effectively suppressing non-radiative transitions of triplet excitons (such as *Angew. Chem. Int. Ed.*, 2022, 61, e202208735; *Adv. Funct. Mater.*, 2024, 2312491).

One paragraph has been added into Page 10, line 187, Revised Manuscript: The

significantly prolonged RTP lifetime of P-Zn glass (124.0 ms) compared with that of P-Zn crystal (6.2 μ s) and P-Cl powder (3.3 μ s) is attributed to the heavy atom effects of Zn and Cl ions, which effectively promote the intersystem crossing process. Additionally, the enhanced density and stiffness of the glass matrix contribute to the formation of a more rigid structure, thus effectively suppressing non-radiative transitions of triplet excitons^{13,31}.

The corresponding references have been added in the Revised Manuscript as below:

13. Zhou, B., Qi, Z. and Yan, D. Highly Efficient and Direct Ultralong All-Phosphorescence from Metal–Organic Framework Photonic Glasses. *Angew. Chem. Int. Ed.* **61**, e202208735 (2022).
31. Gong, Y. et al. Spectral and Temporal Manipulation of Ultralong Phosphorescence Based on Melt-Quenched Glassy Metal–Organic Complexes for Multi-Mode Photonic Functions. *Adv. Funct. Mater.* 2312491 (2024).

3. What is the reason for the photochromic behaviour? Is this a result of ring closure? This is unclear.

Response: Thank you very much for the thoughtful question. As a key branch of photo-responsive materials, photochromism refers to the chemical species capable of reversible color transformation with different absorption spectra under external irradiation (such as *Coordin. Chem. Rev.*, 2022, 452, 214304). The most common processes involved in the photochromism are electron transfers (oxidation–reduction), ring closing/opening, *cis-trans* isomerization, and dissociation (such as *Chem. Soc. Rev.*, 2011, 40, 672).

Recently, considerable effort has been devoted to electron transfer photochromic materials, including viologens, polyoxometalates, spiropyran and naphthalenediimide derivatives (such as *Coordin. Chem. Rev.*, 2023, 475, 214918). In this study, the dopant molecule 4,4'-bipyridine (BP) serves as an electron acceptor, given its widespread use in photochromism, facilitating electron transfer and the generation of BP radicals by combining with suitable electron donors (Cl ions) (such as Nanasawa, M. *Photochromism by Electron Transfer: Photochromic Viologens*, Plenum Press, New

York, 1999, 341–369; *Coordin. Chem. Rev.*, 2019, 378, 533). The photochromic properties of P-Zn-BP glass based on electron transfer has been investigated through a series of experiments and theoretical calculations.

In this work, the photochromism inherent in BP-doped glass has been subjected to comprehensive analysis through UV-vis absorption and electron paramagnetic resonance (EPR) spectra. Upon exposure to 365 nm UV light, the UV-vis spectrum exhibits the emergence of two distinct absorption bands at approximately 400 and 600 nm. The absorption intensity within the 350–700 nm range progressively increases with prolonged irradiation time, as illustrated in Fig. R10a. Concomitantly, the EPR spectrum shows a conspicuous signal characterized by a g value of 2.0016 after irradiation, while no signal is discernible before photoirradiation (Fig. R10b). This observation underscores the pivotal role of light stimulation in generating BP radicals (such as *Angew. Chem. Int. Ed.*, 2022, 61, e202114100; *ACS Appl. Mater. Interfaces*, 2023, 15, 1495), a hypothesis further substantiated through theoretical calculations.

Fig. R10. (a) UV-vis spectra of P-Zn-BP glass at irradiation times of 0, 10, 20, 40, and 60 s. Inset: photographs of P-Zn-BP glass before and after 365 nm UV light irradiations. (b) EPR spectra of P-Zn-BP glass before and after 365 nm light irradiation.

Fig. R10a and Fig. R10b have been added as Fig. 3a and Fig. 3b in the Revised Manuscript, respectively.

Based on density functional theory (DFT) calculations, distribution maps of hole and electron transitions from the ground state (S_0) to the singlet state (S_n) in the P-Zn-BP model reveal evident electron transfer between electron donor Cl anions and electron acceptor BP molecules (Fig. R11). The electron transfer process leads to the

formation of BP radicals. Moreover, the analyses of electrostatic potential (ESP) demonstrate that the blue regions representing more negative ESP values are predominantly concentrated around the Cl anions, highlighting the robust electron-donating capacity of the Cl anions (Fig. R12). This observation aligns with the outcomes from the distribution maps of holes and electrons. Collectively, these findings demonstrate that the electron transfer mechanism should be responsible for the photochromic performance of the BP-doped hybrid glass.

Fig. R11. Distribution maps of the hole and electron of $S_0 \rightarrow S_n$ transition. Green represents the electron-accept part, and the blue means the hole.

Fig. R11 has been added as Supplementary Figure 30 in the Revised Supplementary Information.

Fig. R12. Calculated ESP distribution maps of P-Zn. Bluish colors represent more negative ESP values, and reddish colors mean more positive ESP values.

Fig. R12 has been added as Supplementary Figure 31 in the Revised Supplementary Information.

As suggested from reviewer, some photochromic systems also involve ring closing/open process. The photochromism based on such process after UV irradiation is typically achieved by the derivatives of diarylethenes with/without heterocyclic aryl

groups (Fig. R13) (such as *Chem. Rev.*, 2000, 100, 1685). Upon exposure to UV irradiation, these compounds undergo a transition from a colorless open form featuring a hexatriene core to a closed colored form characterized by an extended π -system with a cyclohexadiene core (such as *Chem. Soc. Rev.*, 2015, 44, 3719). However, unlike these derivatives, the dopant BP does not possess a hexatriene core, and its photochromism does not involve ring-closing reactions.

Fig. R13. Typical photochromic compounds participating in ring-opening/closing reactions.

One paragraph has been added into Page 11, line 214, Revised Manuscript: The photochromism inherent in BP-doped glass was subjected to comprehensive analysis through UV-vis absorption and electron paramagnetic resonance (EPR) spectra. Upon exposure to 365 nm UV light, the UV-vis spectra exhibit the emergence of two distinct absorption bands at approximately 400 and 600 nm. The absorption intensity within the 350–700 nm range progressively increases with prolonged irradiation time, as illustrated in Fig. 3a. Concomitantly, the EPR spectrum shows a conspicuous signal characterized by a g value of 2.0016 after irradiation, while no signal is discernible prior to exposure (Fig. 3b). This observation underscores the pivotal role of light stimulation in generating BP radicals^{33–35}, a hypothesis further substantiated through theoretical calculations (Supplementary Fig. 30).

The corresponding references have been added in the Revised Manuscript as below:

33. Yang, D.-D. et al. Enhancement of Long-Lived Persistent Room-Temperature Phosphorescence and Anion Exchange with Γ^- and SCN^- via Metal–Organic Hybrid Formation. *ACS Appl. Mater. Interfaces* **15**, 1495–1504 (2023).

34. Zhao, G., Liu, W., Yuan, F. and Liu, J. Two Host-Guest 2D MOFs Based on Hexyl Viologen Cations: Photochromism. *Dyes Pigm.* **188**, 109196 (2021).
35. Ding, B., Gao, H., Wang, C. and Ma, X. Reversible room-temperature phosphorescence in response to light stimulation based on a photochromic copolymer. *Chem. Commun.* **57**, 3154–3157 (2021).

One paragraph has been added into Page 13, line 249, Revised Manuscript: Based on density functional theory (DFT) calculations, distribution maps of hole and electron transitions from the ground state (S_0) to the singlet state (S_n) in the P-Zn-BP model reveal evident electron transfer between electron donor Cl anions and electron acceptor BP molecules (Supplementary Fig. 30, 31). The electron transfer process induces the formation of BP radicals, providing a competitive pathway for radiative transition in the form of phosphorescence.

One paragraph has been added into Page 15, line 260, Revised Supplementary Information: In the case of P-Zn OIMHs, the blue regions representing more negative ESP values are predominantly concentrated around the Cl anions, highlighting the robust electron-donating capacity of these Cl anions (Supplementary Fig. 31). This observation aligns with the outcomes from the distribution maps of holes and electrons (Supplementary Fig. 30).

4. The author says that the reversible RTP switching is demonstrated to be repeatable for at least four cycles. What happen after the 4th cycles?

Response: Thanks for the very professional comment. The RTP switching in BP-doped glass is actually controlled by its reversible photochromic property. We have further re-explored the anti-fatigue performance of reversible RTP and discovered that the luminescence switching can be repeated for at least 6 cycles (Fig. R15), highlighting the potential to control persistent luminescence in bulk transparent glass through the photochromic process. Many researches have confirmed that radical ions within electron transfer photochromic compounds are readily oxidized by molecular oxygen to regenerate the starting material or other oxidation products (such as Crano, J. C. and Guglielmetti, R. *Organic Photochromic and Thermochromic Compounds*, Plenum

Press, New York, 1999, 341–343; *Coord. Chem. Rev.*, 2019, 378, 533; *Coordin. Chem. Rev.*, 2019, 378, 533). Consequently, subjecting BP-doped glass to repeated photoirradiation in ambient air may enhance the oxidative degradation of the photogenerated radical products. As a result, after undergoing six cycles of repeated RTP switching, the degree of coloration slightly diminishes under the same irradiation conditions.

Fig. R15. Phosphorescence intensity (at 525 nm) of P-Zn-BP glass after alternating exposure to UV light and heating.

Fig. R15 has been updated as Fig. 3f in the Revised Manuscript.

One paragraph has been added into Page 12, line 233, Revised Manuscript: Crucially, the reversible RTP switching is demonstrated to be repeatable for at least six cycles (Fig. 3f), underscoring the feasibility of modulating persistent luminescence in the bulk transparent glass through the photochromic process. It should be noted that after six cycles of repeated RTP switching, the coloration somewhat diminishes under the same irradiation conditions. This is because subjecting BP-doped glass to repeated photoirradiation in ambient air may enhance the oxidative degradation of the photogenerated radical products³⁶.

The corresponding reference has been added in the Revised Manuscript as below:

36. Sun, J.-K., Yang, X.-D., Yang, G.-Y. and Zhang, J. Bipyridinium derivative-based coordination polymers: From synthesis to materials applications. *Coordin. Chem. Rev.* **378**, 533–560 (2019).

5. From Figure 3e, there seems to be multiple decay rates. What are they associated with?

Response: Thanks for your professional question. As widely understood, the phosphorescence lifetime (τ_p) signifies the duration needed for luminous intensity to decline to 1/e times its peak intensity. The τ_p can be determined using the equation: $\tau_p = 1/(k_r + k_{nr} + k_q)$, where k_r and k_{nr} represent the rates of radiative and non-radiative decay of the lowest triplet excited state (T_1), respectively, and k_q denotes the quenching rate of T_1 .

For the electron transfer photochromic system, the self-absorption of photogenerated radical components conflicts with the emission, resulting in the quenching of luminescence intensity (such as *Angew. Chem. Int. Ed.*, 2022, 61, e202114100; *ACS Appl. Mater. Interfaces*, 2023, 15, 1495; *Chem. Eng. J.*, 2023, 466, 143202; *Chem. Commun.*, 2021, 57, 3154). Spectral analyses of BP-doped glass reveal a notable overlap between ultralong RTP emission and the absorption bands spanning from 450 to 700 nm (see Fig. 3a, 3d). This overlap indicates the presence of self-absorption in the photogenerated radical products. As the photoirradiation time increases, the absorption intensity at 600 nm gradually decreases, indicating that heightened levels of self-absorption predominantly contribute to the decrease of RTP intensity and lifetime post-coloration (Fig. 3d, e).

One paragraph has been added into Page 12, line 239, Revised Manuscript: Further investigation of the generation mechanism for tunable RTP and color in BP-doped glass elucidates a significant overlap between ultralong RTP emission and the absorption band spanning from 450 to 700 nm (Fig. 3a, 3d). This overlap suggests the potential occurrence of self-absorption in the photogenerated radical product. As the photoirradiation time increases, the absorption intensity at 600 nm gradually decreases, indicating that heightened levels of self-absorption predominantly contribute to the decrease of RTP intensity and lifetime post-coloration (Fig. 3d, e).

Reviewer #4:

This is an interesting work reporting the synthesis and investigation of a new

photochromic glass. The work appears as well conducted and the manuscript is clear and well written. Overall it appears as an high quality work reporting some new interesting insights and materials that may have some interesting applications. I do miss however, the comparison of the figure of merits of this new system with state of the art photochromic materials. This comparison could also be made as a form of table that will surely enrich the presentations of the work.

Response: Thank you very much for the positive comment and insightful suggestions regarding this work. Up to now, the reported photochromic materials with photo-switchable room-temperature phosphorescence (RTP) have mainly existed in the forms of single crystals, polymers and powders. Based on your suggestion, we have further conducted a comparative analysis between the photochromic ultralong RTP glass and recently reported materials with photo-controllable RTP. The comparison focuses on key parameters including photo-responsive time, recovery time, and RTP lifetime. We have organized them into a table format as suggested (Table R1).

One paragraph has been added into Page 11, line 207, Revised Manuscript: Compared to state-of-the-art photochromic materials (Supplementary Table 1), P-Zn-BP glass exhibits relatively short photo-responsive time (60 s) and fast recovery time (3 min), while maintaining a long RTP lifetime (107.9 ms). These features underscore the exceptional dynamic ultralong phosphorescence in the large-scale OIMHs glass.

Table R1. Recently reported photochromic materials with photocontrollable RTP.

Sample	Component	Photoresponsive time	Recovery time	RTP lifetime	Ref.
Crystal	Complex 1	1.5 min	60 min ^[a]	99.2 ms	18
	{[Zn(bcbpy) _{0.5} (IPA)] •CH ₃ CN} _n	2 min	30 min ^[a]	100.8 ms	19
	{[Zn ₂ (cbbpy) ₂ (IPA) ₂] •4H ₂ O} _n	1 min	60 min ^[a]	88.0 ms	20

	{[Zn(cbbpy)(HBTC)(H ₂ O)]•2H ₂ O} _n	4.5 min	30 min ^[a]	93.1 ms	21
	[Dy ₂ (H ₂ -HEDP) ₃ (H-HEDP)]•H ₃ -TPB•8H ₂ O	20 min	240 min ^[a]	26.2 ms	22
	[Dy ₂ (H ₂ -HEDP) ₄] ₃ •2(H ₃ -TPA)•2(H ₄ -HEDP)•xsolvent	40 min	*	26.6 ms	23
	(H ₄ BDMPy-Br ₂ NDI)•(NMP) ₄ •(HPW ₁₂ O ₄₀)	3 min	–	1.1 ms	24
	(H ₄ BDMPy-NDI)•(NMP) ₄ •(HPW ₁₂ O ₄₀)	0.5 min	A few hours ^[b]	0.3 ms	25
	(H ₄ BDMPy-NDI)•(NMP) ₁₂ •(H ₃ SiW ₁₂ O ₄₀) ₂	3 min	A few hours ^[b]	9.0 ms	25
	(H ₃ -TPB)•[Zn ₆ (H-HEDP)(HEDP) ₃ (H ₂ O) ₂]•5H ₂ O	60 min	*	0.08 ms	26
	(H-TPB)•[Zn ₃ (H-HEDP)(HEDP)(H ₂ O)]•2H ₂ O	60 min	90 min ^[a]	0.32 ms	26
	Complex 1	2 min	240 min ^[b]	40.3 ms	27
	Complex 2	1 min	30 min ^[a]	66.8 ms	27
	Complex 3	1 min	180 min ^[b]	31.4 ms	27
Powder	Polymer C9	20 s	16 min ^[b]	14.3 ms	28
Film	P1	10 min	8 min ^[a]	20.2 ms	29
	P2	10 min	8 min ^[a]	2.1 ms	29
	NDIA/PVA	4 min	–	34.0 ms	30

Glass	P-Zn-BP	1 min	3 min ^[a]	107.9 ms	This work
--------------	---------	-------	----------------------	----------	-----------

Note: 1. The photo-response time was defined here as the time it took for the intensities of new absorption bands of the photochromic samples to reach saturation. 2. “–”: recover time was not mentioned; “*”: the color could not be recovered. 3. [a]: heating; [b]: put in the dark environment.

Table R1 has been added as Supplementary Table 1 in the Revised Supplementary Information.

The corresponding references have been added in the Revised Supplementary Information as below:

18. Yang, D.-D. et al. Enhancement of Long-Lived Persistent Room-Temperature Phosphorescence and Anion Exchange with I⁻ and SCN⁻ via Metal–Organic Hybrid Formation. *ACS Appl. Mater. Interfaces* **15**, 1495–1504 (2023).
19. Xiao, T. et al. A UV and X-ray dual photochromic Zn (II) metal-organic framework based on viologen: Photo-controlled luminescence and temperature-dependent phosphorescence. *Dyes Pigm.* **208**, 110812 (2022).
20. Yang, D.-D. et al. Multistimuli-Responsive Materials Based on Zn(II)-Viologen Coordination Polymers and Their Applications in Inkless Print and Anticounterfeiting. *Inorg. Chem.* **61**, 7513–7522 (2022).
21. Yang, D.-D. et al. Two multifunctional stimuli-responsive materials with room-temperature phosphorescence and their application in multiple dynamic encryption. *Mater. Chem. Front.* **6**, 2709–2717 (2022).
22. Wei, W.-J., Mu, Y., Wei, L., Hu, J.-X. and Wang, G.-M. Two Photochromic Complexes Assembled by a Nonphotochromic Ligand: Photogenerated Radical Enhanced Room-Temperature Phosphorescence. *Inorg. Chem.* **60**, 108–114 (2021).
23. Feng, D.-X. et al. Photochromic Dy-Phosphonate Assembled by a Pyridine Derivative: Synthesis, Structure, and Light-Enhanced Room-Temperature Phosphorescence. *Cryst. Growth Des.* **22**, 5680–5685 (2022).

24. Di, Y.-M., Li, M.-H., You, M.-H., Zhang, S.-Q. and Lin, M.-J. Photochromic and Room Temperature Phosphorescent Donor–Acceptor Hybrid Crystals Regulated by Core-Substituted Naphthalenediimides. *Inorg. Chem.* **60**, 16233–16240 (2021).
25. Di, Y.-M., Li, M.-H., Zhang, S.-Q., You, M.-H. and Lin, M.-J. Photochromic and Room-Temperature Phosphorescent D–A Hybrid Crystals Induced by Anion– π Interactions. *Cryst. Growth Des.* **21**, 3511–3520 (2021).
26. Feng, D.-X. et al. Light-Induced Electron Transfer Toward On/Off Room Temperature Phosphorescence in Two Photochromic Coordination Polymers. *Adv. Funct. Mater.* **33**, 2305796 (2023).
27. Yang, D.-D. et al. A series of zinc coordination compounds showing persistent luminescence and reversible photochromic properties via charge transfer. *Chem. Eng. J.* **466**, 143202 (2023).
28. Ding, B., Gao, H., Wang, C. and Ma, X. Reversible room-temperature phosphorescence in response to light stimulation based on a photochromic copolymer. *Chem. Commun.* **57**, 3154–3157 (2021).
29. Li, Y., Gu, F., Ding, B., Zou, L. and Ma, X. Photo-controllable room-temperature phosphorescence of organic photochromic polymers based on hexaarylbiimidazole. *Sci. China Chem.* **64**, 1297–1301 (2021).
30. Yao, X. et al. Dynamic room-temperature phosphorescence by reversible transformation of photo-induced free radicals. *Sci. China Chem.* **65**, 1538–1543 (2022).

Thanks again for all the reviewers' valuable suggestion, which undoubtedly enriches our research endeavors.

Reviewer #1 (Remarks to the Author):

The authors revised the manuscript following my previous comments. The current version is ready to be accepted for this journal.

Reviewer #2 (Remarks to the Author):

The author answered all the questions I raised appropriately. I recommend publishing this manuscript in Nature Communications. Additionally, I have been asked to assess the responses to the comments of reviewer #3. The author also solved all the questions raised by reviewer #3.

Reviewer #4 (Remarks to the Author):

The authors have done a very good job answering to all the reviewer questions. I'm in favour of the acceptance of this work in its present form.

Reviewer comments:

Reviewer #1:

The authors revised the manuscript following my previous comments. The current version is ready to be accepted for this journal.

Response: We greatly appreciate the positive comments.

Reviewer #2:

The author answered all the questions I raised appropriately. I recommend publishing this manuscript in Nature Communications. Additionally, I have been asked to assess the responses to the comments of reviewer #3. The author also solved all the questions raised by reviewer #3.

Response: Thank you very much for the positive comment and evaluation of this work.

Reviewer #4:

The authors have done a very good job answering to all the reviewer questions. I'm in favour of the acceptance of this work in its present form.

Response: Thank you very much for the positive comment of this work.